# Single-molecule functional anatomy of endogenous HER2-HER3 heterodimers

**Byoungsan Choi[1,2†‡], Minkwon Cha[2†], Gee Sung Eun[1], Dae Hee Lee[3], Seul Lee[3], Muhammad Ehsan[4§], Pil Seok Chae[4], Won Do Heo[5], YongKeun Park[2], Tae-Young Yoon[1]***

[1]School of Biological Sciences and Institute for Molecular Biology and Genetics, Seoul National University, Seoul, Republic of Korea; [2]Department of Physics, Korea Advanced Institute of Science and Technology (KAIST), Daejeon, Republic of Korea; [3]Proteina Co. Ltd., Seoul, Republic of Korea; [4]Department of Bionanotechnology, Hanyang University, Ansan, Republic of Korea; [5]Department of Biological Sciences, KAIST, Daejeon, Republic of Korea

**Abstract** Human epidermal growth factor receptors (HERs) are the primary targets of many directed cancer therapies. However, the reason a specific dimer of HERs generates a stronger proliferative signal than other permutations remains unclear. Here, we used single-molecule immunoprecipitation to develop a biochemical assay for endogenously-formed, entire HER2-HER3 heterodimers. We observed unexpected, large conformational fluctuations in juxta-membrane and kinase domains of the HER2-HER3 heterodimer. Nevertheless, the individual HER2-HER3 heterodimers catalyze tyrosine phosphorylation at an unusually high rate, while simultaneously interacting with multiple copies of downstream signaling effectors. Our results suggest that the high catalytic rate and multi-tasking capability make a concerted contribution to the strong signaling potency of the HER2-HER3 heterodimers.

*For correspondence:
tyyoon@snu.ac.kr

†These authors contributed equally to this work

Present address: ‡Proteina Co. Ltd., Seoul, South Korea; §Department of Chemistry Mirpur University of Science & Technology, Mirpur, Pakistan

## Introduction

Human epidermal growth factor receptors (HERs) were the first cloned family of receptor tyrosine kinases (RTKs) that provided important insights into the molecular architecture and signaling mechanism of RTKs (*Bargmann et al., 1986*; *Cohen et al., 1982*; *Downward et al., 1984*; *Schechter et al., 1985*; *Yamamoto et al., 1986*; *Yarden and Schlessinger, 1987*). The HER family receptors are frequently overexpressed (*Hendler and Ozanne, 1984*; *Kraus et al., 1987*; *Libermann et al., 1984*; *Slamon et al., 1987*) or mutated (*The Cancer Genome Atlas Network, 2012*; *Lynch et al., 2004*; *Paez et al., 2004*; *Pao et al., 2004*) in many major types of cancers, making them the primary targets of directed cancer therapies. For example, tyrosine kinase inhibitors (TKIs) have been developed that target mutated epidermal growth factor receptors (EGFRs, the first member of the HER family) in lung adenocarcinomas (*Finlay et al., 2014*; *Lynch et al., 2004*; *Paez et al., 2004*; *Pao et al., 2004*; *Remon and Planchard, 2015*; *Stamos et al., 2002*; *Yun et al., 2007*). Similarly, both antibody-based therapies and TKIs have been developed to target HER2 in HER2-overexpressing breast cancers (*Drebin et al., 1985*; *Hudziak et al., 1989*; *Shepard et al., 1991*).

To initiate signaling, HER family receptors assemble with one another to form homodimers (*Garrett et al., 2002*; *Ogiso et al., 2002*; *Yarden and Schlessinger, 1987*), heterodimers (*Goldman et al., 1990*; *Graus-Porta et al., 1997*; *Sliwkowski et al., 1994*; *Wada et al., 1990*), or higher oligomers (*Chung et al., 2010*; *Clayton et al., 2005*; *van Lengerich et al., 2017*; *Zhang et al., 2012*). In particular, HER2-HER3 heterodimers reportedly exhibit the highest transforming capability among all possible combinations of HER family members (*Pinkas-*

*Kramarski et al., 1996*; *Tzahar et al., 1996*). Indeed, elevated levels of HER3 expression correlate with poor prognosis in HER2-positive breast cancer patients (*Junttila et al., 2009*; *Lee-Hoeflich et al., 2008*; *Vaught et al., 2012*).

We anticipate that biochemical assays for the HER family receptors, especially their dimerized forms, will help answer the important question of how a specific combination of HERs can generate stronger proliferative signals than other combinations. If achieved, such a biochemical assay would also serve as a useful screening platform for drugs that inhibit specific types of HER dimer. The traditional route for the construction of such a biochemical assay would be to express and purify individual HERs and then to induce dimerization (or oligomerization) by adding their cognate ligands (*Mi et al., 2008*; *Mi et al., 2011*; *Qiu et al., 2009*). Although many dimeric structures have been reported for truncated parts of the HER family receptors including those of extracellular domains (*Alvarado et al., 2010*; *Cho and Leahy, 2002*; *Cho et al., 2003*; *Freed et al., 2017*; *Garrett et al., 2002*; *Ogiso et al., 2002*), the reconstituted tyrosine (Tyr) kinase activity is marginally low even when attached to vesicle membranes (*Jura et al., 2009*; *Monsey et al., 2010*; *Zhang et al., 2006*), failing to reproduce the strong enzymatic capacity of the HER2-HER3 heterodimers. Reconstitution of full-length HER dimers offers even more experimental challenges, which largely arise from structural instability in their trans- and juxta-membrane domains (*Endres et al., 2013*; *Jura et al., 2009*; *Red Brewer et al., 2009*). Although in vitro dimer assembly has been demonstrated for full-length EGFRs, the enzymatic activity of these in vitro EGFR homodimers decays rapidly with time (*Mi et al., 2008*; *Mi et al., 2011*). Thus, the development of a biochemistry assay for the HER dimers remains a daunting task.

Several groups, including our own, have recently reported the development of single-molecule pull-down and co-immunoprecipitation (co-IP) techniques (*Jain et al., 2011*; *Lee et al., 2018*; *Lee et al., 2013a*; *Yeom et al., 2011*). These experimental methods allow for rapid, non-destructive immobilization of specific cellular complexes on the imaging plane of a single-molecule fluorescence microscope. This makes the immunoprecipitated protein complexes amenable to various biochemical interventions, the results of which can be observed in situ with single-molecule fluorescence imaging. We noted that these unique features of the single-molecule pull-down and co-IP techniques may provide another way to construct a biochemical assay for the HER dimers. In addition, as demonstrated in single-molecule genome sequencing (*Shendure et al., 2005*), single-molecule fluorescence imaging has evolved rapidly toward higher throughput data production. This suggests that a HER dimer biochemical assay, if established via single-molecule pull-down and co-IP, should allow for high-throughput screening of inhibitors for a specific HER dimer pair.

In this study, we sought to establish a biochemical assay for the HER2-HER3 heterodimer. Instead of assembling this HER2 dimer in vitro, we formed HER2-HER3 dimers on live cell membranes with cognate ligands and then rapidly immunoprecipitated these endogenous HER2 dimers on an imaging surface of our single-molecule fluorescence microscope. Strikingly, when we extracted the immunoprecipitated HER2 complexes with cholesterol-like detergents, we found that the HER2 complexes preserved their Tyr kinase activity. We further found that while the HER2-HER3 heterodimer showed unexpected large conformational fluctuations in its cytoplasmic domains, it catalyzed Tyr phosphorylation with an unusually high catalytic rate. In addition, the HER2-HER3 heterodimer can interact with multiple copies of downstream interactors while carrying out Tyr phosphorylation. Our results provide biochemical underpinnings for how the HER2-HER3 heterodimer generates strong proliferative signaling in the physiological context.

## Results

### Single-molecule immunoprecipitation of endogenous HER2-HER3 heterodimers

We first wanted to develop the capability to immunoprecipitate HER2-HER3 heterodimers on our single-molecule fluorescence spectroscope. After adding neuregulin1-beta 1 (NRG1-β1) to HER2-overexpressing SKBR3 cells, we induced IP with antibodies targeting the HER3 extracellular domain (*Figure 1a*). When we used fluorescently-labeled detection antibodies targeting the HER2 cytoplasmic domain, we observed an increase in the immunolabeling count (*Figure 1a and b*). Either omission of NRG1-β1 or addition of NRG1-β1 after the cell lysis step substantially reduced the

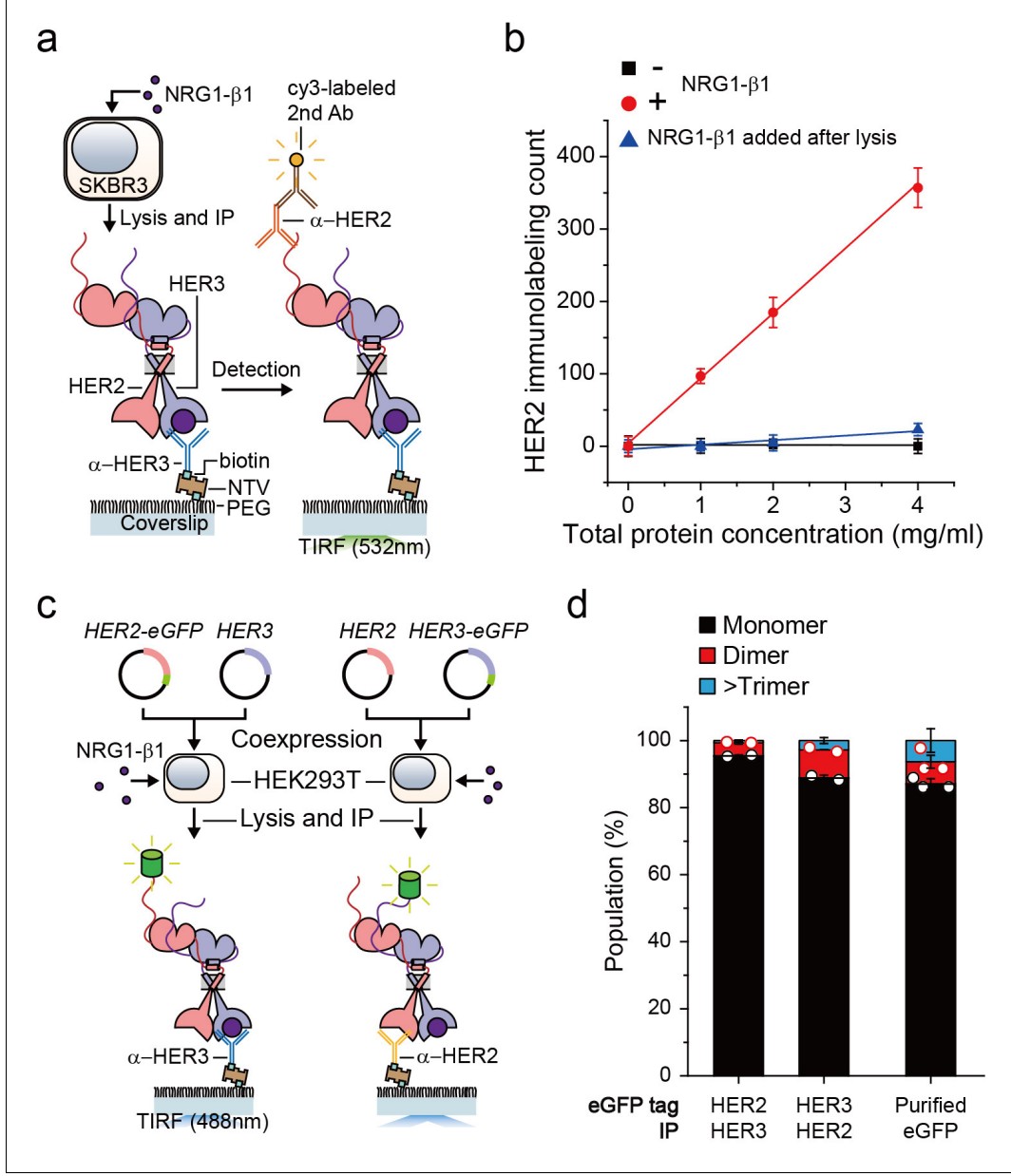

**Figure 1.** Single-molecule IP for HER2-HER3 heterodimers. (**a**) Schematic of single-molecule IP for endogenous HER2-HER3 heterodimers. (**b**) Single-molecule immunolabeling counts of HER2 after HER3 IP from lysates of SKBR3 cells, untreated (square), NRG1-β1 treated during cell culture (circle) or NRG1-β1 treated after lysis (triangle). Data points were obtained from 10 different images (technical replicates; mean, SD) (**c**) Schematic for co-expression and IP of eGFP labeled HER2-HER3 heterodimers within HEK293T. (**d**) Determined the stoichiometry of HER2-HER3 heterocomplexes from HEK293T for indicated eGFP-tagging and IP conditions. HER3-HER2-eGFP heterodimer immunoprecipitated using an α-HER3 antibody (n = 275, 210), HER2-HER3-eGFP heterodimer immunoprecipitated using an α-HER2 antibody (n = 569, 578), eGFP immunoprecipitated using an α-GFP antibody (n = 45, 36, 36; same data were used in *Figure 1—figure supplement 1c* and *Figure 3—figure supplement 1c*). The online version of this article includes the following source data and figure supplement(s) for figure 1:

**Source data 1.** Source data for *Figure 1d*.

**Figure supplement 1.** Stoichiometry and the number of immunoprecipitated HER2-HER3 heterodimer.

number of single-molecule counts (*Figure 1b*). These observations indicated that we pulled down

the endogenous HER2-HER3 hetero-complexes on the surface and that these hetero-complexes formed on the membranes of live SKBR3 cells, not in the cell lysate subsequent to the lysis step.

Using the single-molecule resolution of our fluorescence imaging, we next examined the stoichiometry of HERs in our immunoprecipitated HER2-HER3 hetero-complexes. We developed an isogenic expression system in which we co-expressed HER2 and HER3 in HEK293T cells (*Figure 1c*). We labeled the *HER2* gene with enhanced green fluorescent protein (eGFP), induced HER2-HER3 association with NRG1-β1, and performed single-molecule IP with anti-HER3 antibodies (*Figure 1c*, left). In particular, we confirmed that the HER2-eGFP expression level was equivalent to or even larger than the endogenous HER2 expression observed in the SKBR3 cells (*Figure 1—figure supplement 1a*). When we examined photobleaching of individual complexes, more than 90% of the eGFP spots showed single photobleaching steps, indicating that despite the HER2 overexpression, mainly a single HER2 protein existed in individual HER2-HER3 hetero-complexes (*Figure 1d*; *Figure 1—source data 1*). Next, we labeled the *HER3* gene with eGFP and induced single-molecule IP with anti-HER2 antibodies (*Figure 1c*, right). When pulled down, individual complexes predominantly showed single photobleaching steps as well (*Figure 1d*), which we re-confirmed using the SKBR3 cells (*Figure 1—figure supplement 1b,c*). These data collectively suggest that we mainly immunoprecipitated HER2-HER3 heterodimers with the one-to-one stoichiometry, rather than larger aggregates of HER2 and HER3 proteins.

Finally, for all the surface IP cases we studied, the total counts of the pulled down complexes were kept below 1000 in an imaging area of $40 \times 80\ \mu m^2$, corresponding to a low surface density with a large inter-distance of more than 1.7 μm (on average)(*Figure 1—figure supplement 1d–g*). Thus, despite using bivalent antibodies for our surface IP, we mainly captured a single HER2-HER3 heterodimer per antibody, allowing us to observe mainly single photobleaching steps as in *Figure 1d*. We used this sparse pull-down condition throughout this work unless otherwise specified.

## Immunoprecipitated single HER2-HER3 heterodimers preserve the tyrosine kinase activity

For the HER family receptors, the most critical biochemical process after dimer formation is the generation of phosphorylated tyrosine (pTyr) residues in the C-terminal tails of the receptors. We asked whether our immunoprecipitated HER2 complexes preserve their Tyr kinase activity because this is an important pre-requisite for any further biochemical studies. To this end, we added ATP and Mg$^{2+}$ to the reaction chamber to permit ATP hydrolysis by the immunoprecipitated HER2-HER3 dimers (*Figure 2a*). When we measured pTyr levels using single-molecule immunolabeling with pTyr-specific antibodies, we found increased levels of each of five different pTyr residues in the HER3 tail (HER3 tails have a total of nine Tyr residues) (*Figure 2a and b*; *Figure 2—source data 1*).

Remarkably, as a result of the Tyr phosphorylation reconstituted in HER2-HER3 heterodimers, the pTyr residues in the HER2 tail ere increased as well, by almost same folds as those observed for the HER3 tail (*Figure 2c and d*; *Figure 2—source data 1*). These marked increases in the HER2 pTyr levels essentially required the addition of NRG1-β1 before the cell lysis, indicating that most of the pTyr residues were indeed generated within the HER2-HER3 heterodimers (*Figure 2d*; *Figure 2—source data 1*). In addition, at the low surface density used in our experiments, the majority of the HER2-HER3 heterodimers are biochemically isolated from one another. Thus, the pTyr residues in the HER2 tail should be generated by its own kinase domain, suggesting an active *cis*-phosphorylation activity within a single HER2-HER3 heterodimer.

As we developed our protocol for cell lysis and single-molecule immunoprecipitation, we thus far used a mild cholesterol-like detergent, digitonin. We next asked whether this detergent condition was important for observing the Tyr kinase activity of the immunoprecipitated HER2-HER3 heterodimers. To this end, we examined other mild detergents−CHAPSO, octyl-β-Glucoside (OG), n-dodecyl β-D-maltoside (DDM), Triton X-100 and GDN (*Chae et al., 2012*). We found that all the detergents we tested could be used for the pull-down of HER2-HER3 heterodimers (*Figure 2—figure supplement 1a*). In fact, many of these detergents improved the stability of the immunoprecipitated HER2-HER3 heterodimers over digitonin (*Figure 2—figure supplement 1b,c*). However, when we tested the Tyr phosphorylation in the HER2-HER3 heterodimers, only GDN preserved the tyrosine kinase activity at levels comparable to digitonin (*Figure 2e—source data 1*; *Chae et al., 2012*).

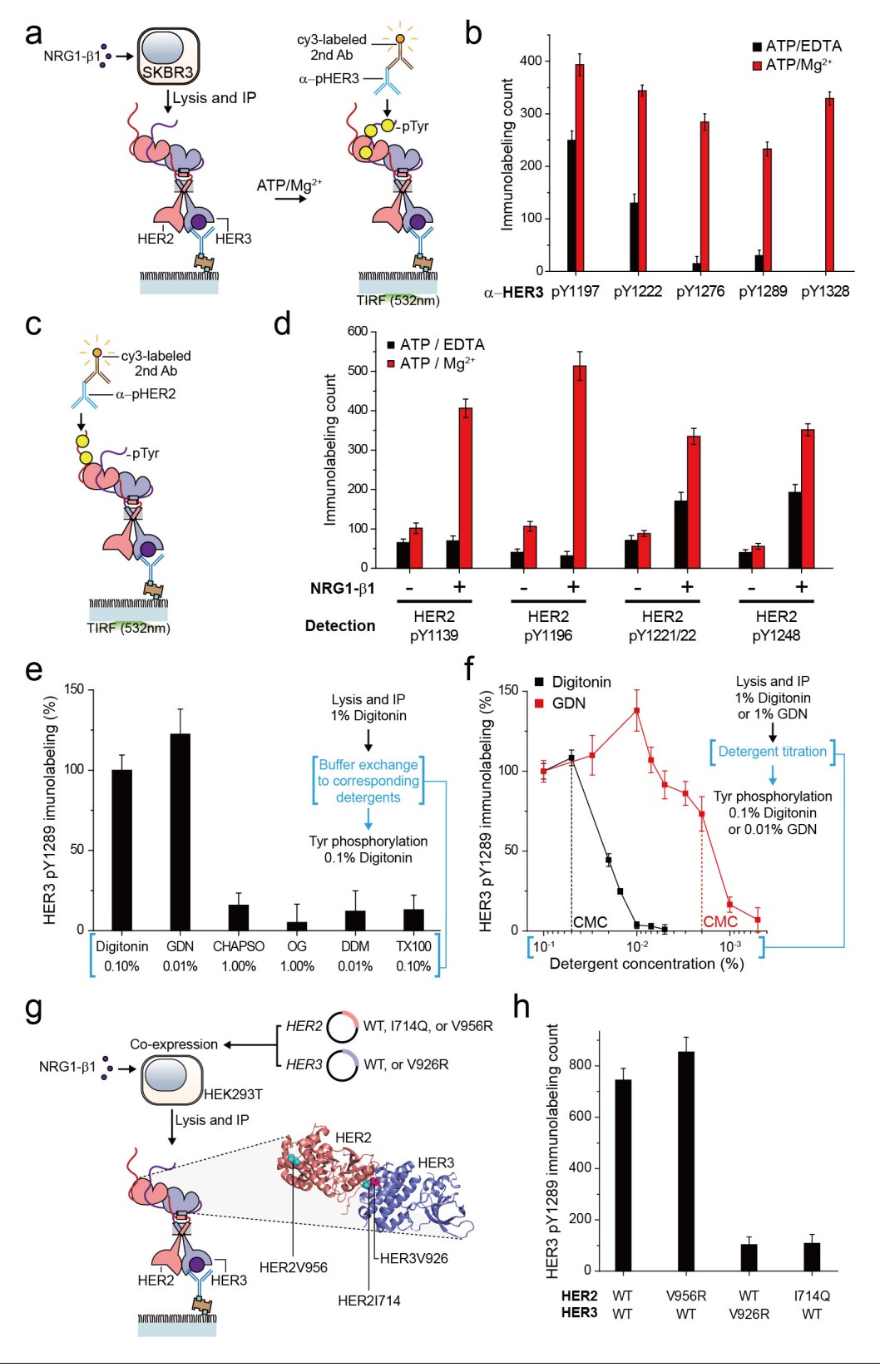

**Figure 2.** Immunoprecipitated single HER2-HER3 heterodimers preserve the tyrosine kinase activity. (**a**) Schematic for the Tyr phosphorylation of immunoprecipitated HER2-HER3 heterodimers. (**b**) Single-molecule immunolabeling counts for specific pTyr residues of the immunoprecipitated HER2-HER3 heterodimers. Prior to surface IP of the
*Figure 2 continued on next page*

*Figure 2 continued*

heterodimers, the heterodimers were dephosphorylated with endogenous tyrosine phosphatases by not including phosphatase inhibitors in the lysis buffer. (**c**) Schematic for the HER2 pTyr immunolabeling after Tyr phosphorylation of HER2-HER3 heterodimers. (**d**) Phosphorylation of HER2 Tyr sites on HER2-HER3 heterodimer. Prior to surface IP of the heterodimers, the heterodimers were dephosphorylated with endogenous tyrosine phosphatases by not including phosphatase inhibitors in the lysis buffer. HER3 was immunoprecipitated from the lysates of SKBR3 cells treated (+) or untreated (-) with NRG1-β1. The number of phosphor-Tyr was measured using indicated pTyr-specific antibodies after phosphorylation. (**e**) Dependence of the Tyr phosphorylation activity of the HER2-HER3 heterodimer on the types of detergents used for functional reconstitution. The phosphorylation level with digitonin is considered as 100% activity (mean, SD). (**f**) Detailed titration of the digitonin and GDN concentrations and its effect on the HER2-HER3 heterodimer Tyr kinase activity. Phosphorylation level at 0.1% for each detergent is considered as 100% activity (mean, SD). (**g**) Schematic for the immunoprecipitation of key point mutation-bearing HER2-HER3 heterodimers. (**h**) Increase of HER3 pTyr1289 level after adding ATP and Mg2 + to HER2-HER3 heterodimers which bear key point mutations. All data points and bar graphs were obtained from 10 different images (technical replicates; mean, SD).

The online version of this article includes the following source data and figure supplement(s) for figure 2:

**Source data 1.** Source data for *Figure 2b, d, e and h*.
**Figure supplement 1.** Immunoprecipitation of HER2-HER3 heterodimer within various detergent species.

---

We next titrated the digitonin and GDN concentrations and studied their effects on the levels of pTyr 1289 in the HER3 tail. Our results clearly indicated that digitonin and GDN should be maintained at levels above their respective critical micellar concentrations (CMCs) to maintain the Tyr kinase activity (*Figure 2f*). We further noted that restoring 0.1% digitonin (or 0.01% GDN) after depletion of digitonin or GDN to levels below their respective CMCs did not rescue the tyrosine kinase activity of the HER2-HER3 heterodimers (*Figure 2f*, third step). Since both digitonin and GDN have structures similar to cholesterol, these results collectively suggest digitonin and GDN provide a membrane-like environment that is essential for maintaining the Tyr kinase activity of the immunoprecipitated HER2-HER3 heterodimers.

In activating their Tyr kinase activity, the HER family receptors form an asymmetric dimer in their kinase domains (*Jura et al., 2009*; *Red Brewer et al., 2009*; *Zhang et al., 2006*), which distinguishes HERs from other RTK families in terms of the activation mechanism (*Lemmon et al., 2014*). We asked whether the immunoprecipitated HER2-HER3 heterodimers use the asymmetric dimer for their Tyr kinase activity. We adopted the isogenic expression system with HEK293T cells (*Figure 2g*), which we used to determine the single-molecule stoichiometry of the HER2-HER3 heterodimers. We first introduced mutations that impaired the activator function of either HER2 (V956R) or HER3 (V926R). Only the activator mutation introduced to HER3, not HER2, ablated the Tyr kinase activity of the resulting HER2-HER3 heterodimers (*Figure 2h—source data 1*). On the other hand, when we introduced a receiver mutation to HER2 (I714Q), we observed an impaired dimer kinase activity (there is no corresponding receiver mutation for HER3) (*Figure 2h—source data 1*). These results indicate that the immunoprecipitated HER2-HER3 heterodimers rigorously use the asymmetric dimer configuration for reconstituting their Tyr kinase activity. In this configuration, the HER3 kinase predominantly played the role of the activator and the HER2 kinase acted mainly as the receiver.

## Single-molecule kinetic and conformational analysis of the immunoprecipitated HER2-HER3 heterodimer

Based on our established assay, we sought to study the biochemical mechanisms underlying the strong signaling potency of the HER2-HER3 heterodimer. To this end, we characterized the enzymatic kinetics of single HER2-HER3 heterodimers. We immunoprecipitated endogenous HER2-HER3 heterodimers from SKBR3 cell membranes and measured the increase in pTyr 1289 level (in the HER3 tail) as a function of increasing reaction time (*Figure 3a*). Because we experimentally confirmed the linearity of our single-molecule immunolabeling for pTyr1289 (*Figure 3—figure supplement 1a,b*), our determined increase in pTyr1289 could be directly converted to a rate of the corresponding Tyr phosphorylation. By repeating these experiments with increasing concentrations of ATP, we obtained a Michaelis-Menten curve associated with the Tyr 1289 phosphorylation by single HER2-HER3 heterodimers (*Figure 3b and c*; *Figure 3—source data 1*). We found a Michaelis

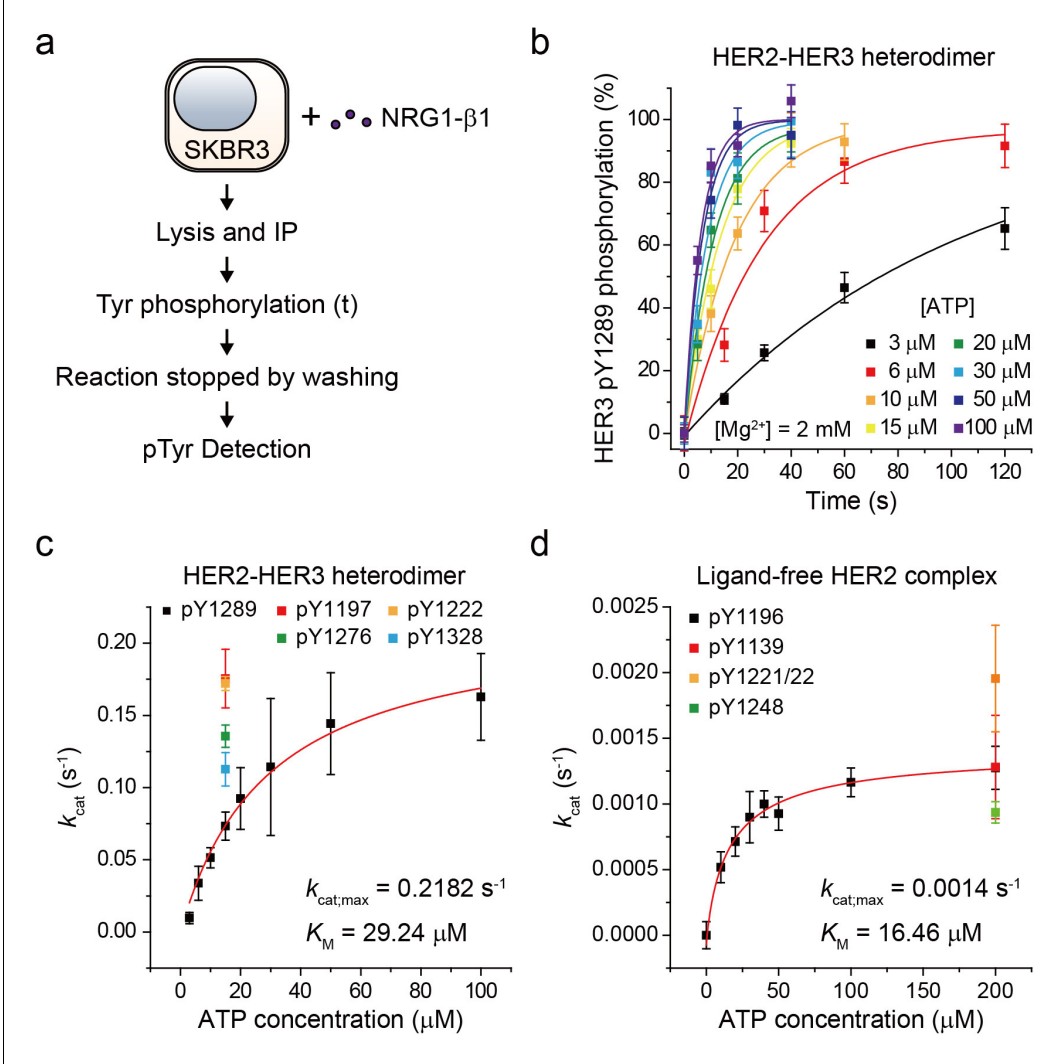

**Figure 3.** Single-molecule kinetic and conformational analysis of the immunoprecipitated HER2-HER3 heterodimer. (**a**) Schematic for catalytic rate measurements of HER2-HER3 heterodimer (**b**) The amount of HER3 pTyr1289 measured as an increasing reaction time for phosphorylation with indicated concentration of ATP and $Mg^{2+}$. Each curve was normalized by the expected maximum number of pTyr1289 calculated by extrapolating an exponential fit. (**c**) Michaelis-Menten curve for HER2-HER3 heterodimers. Each catalytic rate was obtained from the measurements in (b). At 15 µM ATP, the catalytic rates for HER3 pTyr1197 (red), 1222 (orange), 1276 (green), and 1328 (blue) were independently measured. (**d**) Michaelis-Menten curve for ligand-free HER2 complexes. All data points were obtained from 10 different images (technical replicates; mean, SD).

The online version of this article includes the following source data and figure supplement(s) for figure 3:

**Source data 1.** Source data for *Figure 3c and d*.

**Figure supplement 1.** Linearity of pTyr immunolabeling and stoichiometry of HER2-eGFP ligand free complex.

constant ($K_M$) of 29.2 µM and a maximum catalytic rate ($k_{cat,max}$) of $2.2 \times 10^{-1}$ $s^{-1}$ (*Figure 3c*; *Figure 3—source data 1*). We measured the kinetic rates for phosphorylation at other tyrosine residues and found that all the catalytic rates were of a similar order of magnitude around ~0.1 $s^{-1}$ when measured at 15 µM [ATP] (*Figure 3c*; *Figure 3—source data 1*).

To have a quantitative comparison for the determined enzymatic parameters, we used the approach developed above to construct a biochemical assay for ligand-free HER2 complexes. We confirmed immunoprecipitation of ligand-free HER2 complexes from the isogenic expression system with HEK293T cells, where we overexpressed eGFP-labeled HER2 proteins. Single-molecule photobleaching analysis revealed that the majority of the immunoprecipitated ligand-free HER2 complexes

included two HER2 proteins, but a non-negligible fraction of complexes involved more than three HER2 proteins (*Figure 3—figure supplement 1c*). Thus, the stoichiometry is less strictly defined in this ligand-free case (this is why we refer to these immunoprecipitates as ligand-free HER2 complexes, not dimers).

As done for the HER2-HER3 heterodimer, we added ATP and Mg$^{2+}$ to the ligand-free HER2 complexes (immunoprecipitated from the SKBR3 cell membranes) and observed increases in the pTyr residues in the HER2 tails, indicative of the reconstituted Tyr kinase activity. By titrating the ATP concentration during the reconstituted Tyr kinase reaction, we obtained the Michaelis-Menten curve for the Tyr 1196 phosphorylation by single ligand-free HER2 complexes. We obtained a $K_M$ of 16.46 μM and a $k_{cat,max}$ of $1.4 \times 10^{-3}$ s$^{-1}$, respectively, for this specific Tyr phosphorylation (*Figure 3d*; *Figure 3—source data 1*). Remarkably, we found that while $K_M$ of single HER2-HER3 heterodimers falls in a similar range to that of the ligand-free HER2 complexes (29.2 μM versus 16.5 μM), $k_{cat,max}$ of the HER2-HER3 heterodimers is 150 times higher than that of the ligand-free HER2 complexes. Together, our results suggest that it is $k_{cat,max}$, rather than $K_M$, that distinguished the HER2-HER3 dimers from the ligand-free HER2 complexes.

Next, to probe the conformations of the two kinase domains, we combined the single-molecule fluorescence resonance energy transfer (FRET) and the single-molecule co-IP imaging (*Figure 4a*). In our isogenic HEK293T cell expression system, we introduced an 11-residue sequence, YbbR, between juxta-membrane domain A (JM-A) and JM-B in the HER2 and HER3 proteins, respectively (*Figure 4a*). We examined different locations for insertion of the YbbR sequence and found that insertion after residue 694 (for HER2) and 684 (for HER3) with a flanking linker sequence maintained the Tyr kinase activity of the HER2-HER3 heterodimers (~40% compared with the wild-type (WT) dimers) (*Figure 4b* and *Figure 4—figure supplement 1a*). For the EGFR dimers, asymmetric dimer formation has been shown to strictly require an anti-parallel coiled-coil structure for the JM-As (*Jura et al., 2009*; *Red Brewer et al., 2009*). In our case, this should bring the two respective YbbR sequences into close proximity. Thus, when we observed low FRET efficiency (*E*), we were able to assume JM-A dissociation and thus absence of the asymmetric dimer structure, and vice versa (*Figure 4a*, bottom inset). After cell lysis, we added Sfp phosphopantetheinyl transferase with its substrates Cy3-CoA and Cy5-CoA to the cell extract, which led to the specific labeling of the Cy3 and Cy5 dyes on the YbbR sequences (*Figure 4—figure supplement 1b,c*; *Yin et al., 2006*). After immunoprecipitation, we collected single-molecule FRET signals only from the diffraction-limited fluorescence spots that were properly labeled with a single pair of Cy3 and Cy5 dyes.

In *Figure 4c*, we plotted the *E* value of individual ligand-free HER2 complexes. We set a threshold at an *E* value of 0.5, which represents a separation of ~5 nm between the labeled Cy3 and Cy5 dyes (we also observed local minima at *E* = 0.5). We found that a broad *E* distribution for the ligand-free HER2 complexes with about 50% of populations showing *E* values lower than 0.5. This observation indicated significant conformational fluctuations in their kinase domains (*Figure 4c*, inset). When examining time-resolved traces, we indeed observed transitions between high and low FRET states for a portion of single ligand-free HER2 complexes, while other complexes remained stable in either the high or low FRET state (*Figure 4d* and *Figure 4—figure supplement 1d*). Remarkably, when we examined the FRET efficiency of single HER2-HER3 heterodimers, we found a comparable population (35%) with *E* values lower than 0.5 (*Figure 4e*, black symbol and inset). In addition, we found a fraction of individual HER2-HER3 heterodimers that showed repetitive shuttling between the high and low FRET states (*Figure 4f* and *Figure 4—figure supplement 1e*).

Finally, we added HER2-targeting TKI, lapatinib, to our immunoprecipitated HER2-HER3 heterodimers. Lapatinib is known to selectively bind to non-active conformation of the HER kinase domains (*Wood et al., 2004*). Indeed, we found that addition of lapatinib before ATP hydrolysis buffer significantly increased the low FRET populations (*E* < 0.5) for both HER2-HER3 heterodimers and ligand-free HER2 complexes (*Figure 4c and e*, red versus black distributions). We note the possibility that different FRET populations, especially, those with low *E* values, reflect some degraded HER2 dimers or complexes. In addition, it is possible that solubilization of the HER2 dimers in micelles renders the low FRET populations (with JM-As dissociated) overestimated compared with the physiological populations in the cellular membranes. However, the FRET changes in response to the lapatinib incubation per se report detailed conformational changes in the single HER2 dimers. They also suggest that binding of lapatinib to the non-active conformation of the HER2 kinase domains leads to total dissociation of the asymmetric dimer of kinase domains. Altogether, our data suggest that the

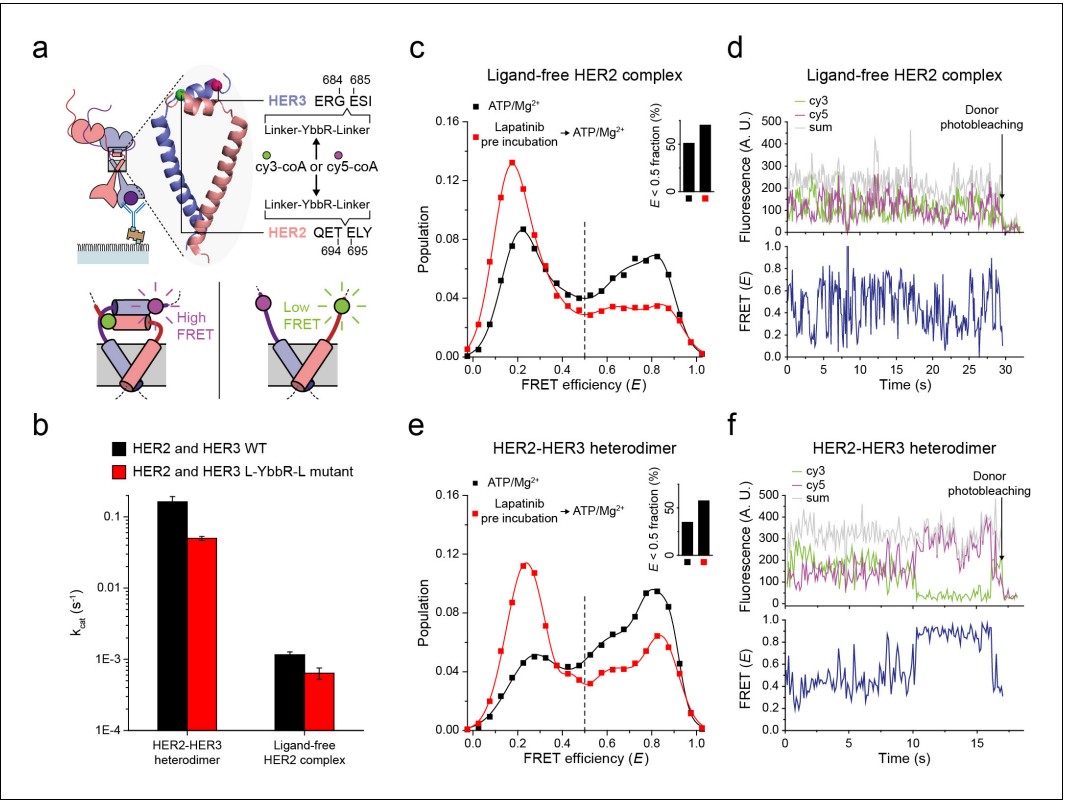

**Figure 4.** Structural investigation of HER2 complexes using single-molecule FRET. (**a**) Schematic for labeling HER2-HER3 heterodimers with fluorescent dyes. For specific labeling, a YbbR sequence was inserted between juxta-membrane-A and juxta-membrane B with an appropriate linker. Cy3 or Cy5 was randomly attached to the YbbR sequence afterward. (**b**) The catalytic rates of the WT and YbbR-inserted HER2-HER3 heterodimer and ligand-free HER2 complex) (fitting result, SE). (**c**) Distribution of FRET efficiency from fluorescence-labeled ligand-free HER2 complex. The ligand-free complexes were either untreated (black, n = 208) or pre-treated with lapatinib (red, n = 231) before being imaged after adding ATP and Mg$^{2+}$. (**d**) Fluorescence trace from ligand-free HER2 complex labeled with both Cy3 and Cy5. (**e**) Distribution of FRET efficiency from fluorescence-labeled HER2-HER3 heterodimer. The heterodimers were either untreated (black, n = 281) or pre-treated with lapatinib (red, n = 218) before being imaged after adding ATP and Mg$^{2+}$. (**f**) Fluorescence trace from HER2-HER3 heterodimers labeled with both Cy3 and Cy5. See Materials and methods.

The online version of this article includes the following figure supplement(s) for figure 4:

**Figure supplement 1.** Catalytic rate of YbbR tagged HER2 complexes and their representative fluorescence traces.

cytoplasmic domains of single HER2-HER3 heterodimers show significant conformational dynamics at an almost equal level to those of the ligand-free HER2 complexes.

## Single HER2-HER3 heterodimers interact with multiple copies of downstream interactors

We next asked whether this increase in pTyr led to an actual increase in interactions with downstream interactors. We used our single-molecule co-IP imaging technique, wherein we examine protein-protein interactions (PPIs) between the immunoprecipitated cellular complexes and eGFP-labeled prey proteins (*Figure 5a*; *Lee et al., 2018*; *Lee et al., 2013a*). We transiently expressed prey proteins tagged with eGFPs in HEK293T cells and added a diluted lysate of these HEK293T cells (eGFP-labeled preys at 10 nM) to initiate the corresponding PPIs. This single-molecule co-IP imaging can be viewed as a reconstitution of downstream PPIs using the exogenously prepared prey proteins, which has a net effect of amplifying PPIs to increase the readout and dynamic range of the co-IP analysis (*Figure 5a* and *Figure 5—figure supplement 1a*).

To study the stoichiometry of these PPIs, we determined the number and corresponding eGFP fluorescence intensity of the prey proteins that simultaneously interacted with individual HER2-HER3 heterodimers (*Figure 5b and c*). When we added tandem SH2 domains of phospholipase C gamma (PLCγ$_{SH2}$) in the absence of in vitro Tyr phosphorylation, we found that mainly, one or two PLCγ$_{SH2}$ interacted with single HER2-HER3 heterodimers (*Figure 5b and d*). Remarkably, reconstituted Tyr phosphorylation broadened the fluorescence intensity and number distributions, indicating many more copies of PLCγ$_{SH2}$ simultaneously bound to single HER2-HER3 heterodimers (*Figure 5b and d*).

We also examined the PPIs between HER2-HER3 dimers and eGFP-labeled p85α—a regulatory subunit of phosphatidylinositol-4,5-bisphosphate 3-kinase (PI3K). There are seven known binding sites for PI3K per HER2-HER3 dimer (five on HER3 and two on HER2) (*Hause et al., 2012*; *Jones et al., 2006*). Despite the larger size of p85α (i.e., 85 kDa versus 23 kDa of PLCγ$_{SH2}$), we found multiple p85αs interacting with each HER2-HER3 heterodimer after reconstituted Tyr phosphorylation (*Figure 5c and e*). Antibody blocking experiments with pTyr-specific antibodies showed that different pTyr residues in the HER3 tail had differentiated contributions to the observed PPIs (*Figure 5—figure supplement 1b–d*). We also induced assembly of the full PI3K complex through co-expression of mCherry-labeled p85α and the eGFP-labeled catalytic p110 subunit (α isoform) in the same group of HEK293T cells. We found multiple copies of these full PI3Ks interact with single HER2-HER3 heterodimers (*Figure 5f*). Together, these observations indicate that individual HER2-HER3 heterodimers can physically interact with multiple copies of downstream interactors simultaneously.

We noted that the single-molecule resolution of our co-IP imaging might provide a chance to answer the long-standing question of whether there is any specific order in the effector binding to the HER oligomers. To develop an insight into this question, we introduced the tyrosine-to-phenylalanine (Y-to-F) mutation to six different Tyr residues in the HER3 tail using our isogenic expression system (*Figure 5g*). We reasoned that if there were any patterns dictating the orders of effector binding, we would observe marked drops in the effector binding in a certain set of mutants. This is because with no phosphorylation in the specific Tyr residue, any further bindings occurring in a pre-defined order (i.e., the pattern in the effector binding) would be prevented.

We collected time-resolved traces of p85α binding for the six mutants (total ~25,000 traces), and identified the number of p85α binding for individual traces as we did in *Figure 5c*. When examining the distributions of the p85α binding counts, we failed to observe any such marked changes in the p85α binding counts across the six Y-to-F mutants we studied (*Figure 5h–m*). All the mutants showed only slight decreases in the p85α binding counts, with virtually every peak being moved to exactly one less binding. These data suggest that the effector bindings in our in vitro setting largely occur in a stochastic manner without a specific pattern in their binding order.

## Single HER2-HER3 heterodimers simultaneously carry out Tyr phosphorylation and downstream PPIs

So far, we examined reconstitution of the Tyr kinase activity and the PPIs with downstream interactors for single HER2-HER3 heterodimers. We next asked whether a single HER2-HER3 heterodimer can carry out these two functions simultaneously. We noted that activation of the Tyr kinase activity of the HER family receptors has many conformational requirements, including asymmetric dimerization of the Tyr kinase domains as well as the anti-parallel coiled-coil structure of the juxta-membrane domains, as we examined above in *Figures 2* and *4* (*Jura et al., 2009*; *Red Brewer et al., 2009*). In addition, the C-terminal tails allegedly wrap around the asymmetric dimer of the kinase domains (*Kovacs et al., 2015*). There is thus an ample possibility that the conformations required for Tyr phosphorylation are incommensurate with PPIs with downstream interactors. We wondered whether major conformational changes are required subsequent to Tyr phosphorylation to permit PPIs with downstream proteins, or vice versa (*Figure 6a*, Model I).

To answer this question, we introduced purified tyrosine phosphatase PTPN1 at a saturating concentration of 10 µg/ml to our Tyr phosphorylation process (*Figure 6b*). This excess PTPN1 created a dynamic equilibrium between Tyr phosphorylation and dephosphorylation for single HER2-HER3 heterodimers. With the reaction conditions we used, a single pTyr 1289 residue was generated and removed with kinetic rates of 0.8 s$^{-1}$ and 0.58 s$^{-1}$, respectively (*Figure 6—figure supplement 1a–c*), indicating that multiple rounds of Tyr phosphorylation occurred within each HER2-HER3 dimer

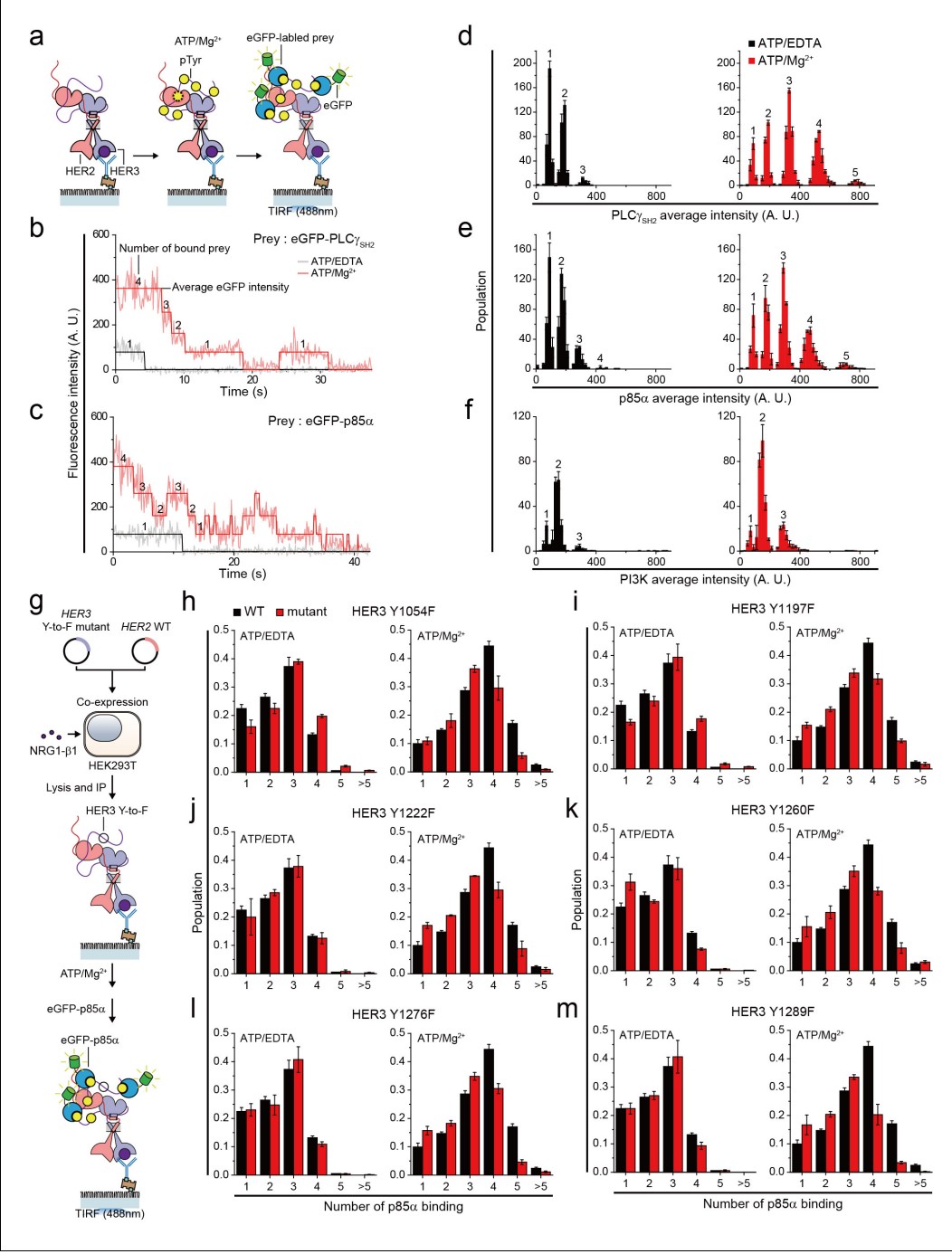

**Figure 5.** Single HER2-HER3 heterodimers interact with multiple copies of downstream interactors. (**a**) Schematic for single-molecule co-IP imaging after reconstituted HER2-HER3 heterodimer phosphorylation. (**b, c**) Fluorescence traces of single PPI complexes between eGFP-labeled preys (PLCγ$_{SH2}$ or p85α) and phosphorylated (red) or unphosphorylated (black) HER2-HER3 heterodimers. Number of eGFPs within each PPI complex is indicated. (**d–f**) Distribution of average intensity from the maximum step (see b) from PPI complexes consisting of eGFP-labeled preys (PLCγ$_{SH2}$, p85a, or PI3K) and HER2-HER3 heterodimers. Counted number of eGFP-preys bound to HER2-HER3 heterodimers is indicated at the top of peak. The result is average of 3 distribution group and each group consists of hundreds traces (PLCγ$_{SH2}$, ATP+EDTA; n = 612, 651, 616, ATD+Mg$^{2+}$; n = 1087, 1085, 1046/p85α, ATP+EDTA; n = 649, 592, 695, ATD+Mg$^{2+}$; n = 899, 910, 871/PI3K, ATP+EDTA; n = 481, 476, 529, ATD +Mg$^{2+}$; n = 726, 782, 863). Population was calibrated by used concentration of SKBR3 lysate (technical replicates; mean, SD). (**g**) Schematics for HER2-HER3 heterodimers which bear indicated Tyr to Phe point mutations. (**h–m**)
*Figure 5 continued on next page*

*Figure 5 continued*

Population of the maximum step from PPI complexes consisting of eGFP-labeled p85α and HER2-HER3 heterodimers which bear indicated point mutations. The result is average of 3 distribution group and each group consists of hundreds traces (Y1054F, ATP+EDTA; n = 649, 592, 695, ATP+Mg$^{2+}$; n = 899, 910, 871/Y1197F, ATP +EDTA; n = 649, 592, 695, ATP+Mg$^{2+}$; n = 899, 910, 871/Y1222F, ATP+EDTA; n = 649, 592, 695, ATP+Mg$^{2+}$; n = 899, 910, 871/Y1260F, ATP+EDTA; n = 649, 592, 695, ATP+Mg$^{2+}$; n = 899, 910, 871/Y1276F, ATP+EDTA; n = 649, 592, 695, ATP+Mg$^{2+}$; n = 899, 910, 871/Y1289F, ATP+EDTA; n = 649, 592, 695, ATP+Mg$^{2+}$; n = 899, 910, 871). Population was calibrated by average number of traces for each ATP+EDTA condition (technical replicates; mean, SD).

The online version of this article includes the following figure supplement(s) for figure 5:

**Figure supplement 1.** p85α immunolabeling and PPI inhibition by pTyr antibody binding.

during the given 3 min observation time. A larger Mg$^{2+}$ concentration of 10 mM was used to obtain a higher phosphorylation rate than in *Figure 3*. In addition, instead of examining the p85α binding subsequent to Tyr phosphorylation (e.g., in *Figure 5*), we examined in situ binding of p85α (*Figure 6b*).

We reasoned that if two separate conformations were required for Tyr phosphorylation and for PPIs with eGFP-labeled preys, many, if not all, of the bound interactors would need to be removed before a new round of Tyr phosphorylation could occur (*Figure 6a*, red arrow of Model I). As we observed in *Figure 5e*, the fluorescence intensity distribution of eGFP-p85α PPI spots becomes much broader in the presence of ATP hydrolysis buffer, indicative of the binding of multiple copies of p85α (*Figure 6c*, black versus blue distributions). Remarkably, we found that this broad fluorescence intensity distribution is not much changed even after addition of at 10 μg/ml PTPN1 (*Figure 6c*, blue versus red distributions). These observations indicate multiple copies of eGFP-p85α still bound to single HER2-HER3 dimers in the presence of excess PTPN1.

When examining the corresponding time-resolved traces, we found that recurrent binding and unbinding events of eGFP-p85α did occur for each HER2-HER3 dimer (*Figure 6d*). Using hidden Markov modeling, we measured the time gaps between the first p85α binding and any additional binding events (Δt). If these binding events were all due to the pTyr residues generated in the same round of Tyr phosphorylation (thus following Model I of *Figure 6a*), the Δt distribution would decrease monotonically with a similar time constant of Tyr dephosphorylation (e.g., *Figure 6—figure supplement 1c*). In contrast, we observed continuous binding events with a long tail that extended to tens of seconds (*Figure 6e*), strongly suggesting that each HER2-HER3 heterodimer generates new pTyr residues while still interacting with more than one eGFP-p85α prey protein. Finally, we examined PPIs of the ligand-free HER2 complexes under the same concentrations of ATP, Mg$^{2+}$ and PTPN1. We found that unlike the case of the HER2-HER3 heterodimers, the fluorescence intensity distribution substantially shrank down upon addition of PTPN1 (*Figure 6f*, blue versus red distributions). Together, out data point to a multi-tasking capability unique to the HER2-HER3 dimer that permits both pTyr phosphorylation and PPIs with downstream interactors in a simultaneous manner.

## The high catalytic rate of the HER2-HER3 heterodimer is pivotal to its lapatinib resistance

Our observation suggests that the single HER2-HER3 heterodimers continue to generate pTyr residues and attract downstream interactors even under harsh dephosphorylating conditions. Our observations made in *Figure 3* indicate that it is the high catalytic rate that mostly distinguishes the HER2-HER3 heterodimer from the ligand-free HER2 complexes, the latter of which almost completely lose their function under the same dephosphorylating conditions. We thus wondered whether the high catalytic rate of the HER2-HER3 heterodimer is indeed pivotal to the persistent Tyr kinase activity and downstream effector PPIs.

To gauge the signaling strength of the single HER2-HER3 heterodimers in a more quantitative manner, we added lapatinib to the reconstituted Tyr kinase reaction of the HER2-HER3 heterodimer and measured the degree of inhibition (*Figure 7a*). This yielded an inhibition curve with an IC$_{50}$ of 6 μM for the single HER2-HER3 heterodimers (*Figure 7b*). On the contrary, the Tyr kinase activity of the ligand-free HER2 complexes was observed to be more readily quenched by lapatinib with an IC$_{50}$ of 0.7 μM, a value one order of magnitude smaller than determined for the HER2-HER3

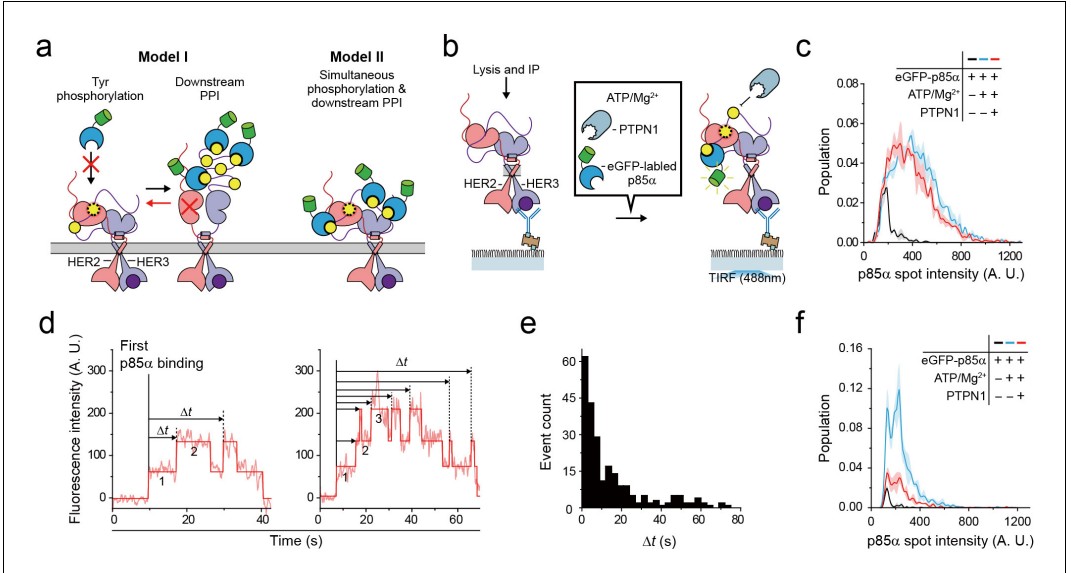

**Figure 6.** Single HER2-HER3 heterodimers simultaneously carry out Tyr phosphorylation and downstream PPIs. (a) Two models. Model I: PPIs permitted after tyrosine phosphorylation. Model II: both processes occur simultaneously. (b) Schematic for the all-in-one reaction. PTPN1, eGFP-p85α, and ATP / Mg$^{2+}$ were added to immunoprecipitated HER2-HER3 heterodimers. (c) Distribution of fluorescence intensity from single PPI complexes with eGFP-p85α under the indicated conditions (HER2-HER3 heterodimers). Images were recorded from the injection of reactants (i.e., eGFP-p85α, ATP, Mg$^{2+}$, and PTPN1). Maximum intensities were collected from individual fluorescence traces (black; n = 156, 155, 199/blue; n = 1139, 1187, 1169/red; n = 991, 1029, 1064). (d) Fluorescence traces from the all-in-one reaction. The number of bound eGFP-p85α are indicated. Dwell times from the first binding of eGFP-p85α (indicated vertical line) to additional binding events (dotted vertical lines) were collected until there were no remaining eGFP-p85α bound to HER2-HER3 heterodimers (until photobleaching). (e) Population of collected dwell time at (d). n = 34 individual single-molecule traces were used. (f) Distribution of fluorescence intensity from PPI complexes with eGFP-p85α under the indicated conditions (ligand-free HER2 complexes). Images were recorded from the injection of reactants (black; n = 41, 36, 39/blue; n = 648, 753, 788/red; n = 233, 205, 227).

The online version of this article includes the following figure supplement(s) for figure 6:

**Figure supplement 1.** Catalytic rate of HER2-HER3 heterodimer and PTPN1.

heterodimer. This discrepancy in the IC50 values is consistent with the known, strong resistance of the HER2-HER3 heterodimer against lapatinib inhibition (*Novotny et al., 2016*; *Wilson et al., 2012*).

As we did in *Figure 6*, we next added purified Tyr phosphatases, PTPN1, to generate an equilibrium between Tyr phosphorylation and dephosphorylation (*Figure 7a*). We reasoned that PTPN1 should reduce the effective speed of Tyr phosphorylation (thus the effective catalytic rate) without directly affecting the conformational changes of HER2-HER3 heterodimers observed in *Figure 4*. Remarkably, the presence of PTPN1 markedly altered the lapatinib inhibition curve, shifting the curve to lower lapatinib concentrations by almost two orders of magnitude at the highest PTPN1 concentration we used (*Figure 7b*). Our observation suggests that the effective catalytic rate reduced by PTPN1 makes the HER2-HER3 heterodimers more vulnerable to lapatinib inhibition. Indeed, in the case of the HER2-free ligand complexes that have a much lower catalytic rate, the addition of PTPN1 alone steeply reduces Tyr phosphorylation activity of (*Figure 7c*). Together, our results suggest the high catalytic rate of the HER2-HER3 heterodimers is a key biochemical feature endowing the HER2-HER3 heterodimers with the persistent signaling activity as well as the lapatinib resistance.

## Discussion

In this study, we explored a novel route to the construction of a biochemical assay of HER2-HER3 heterodimer function. Instead of attempting in vitro assembly, we formed the HER2-HER3

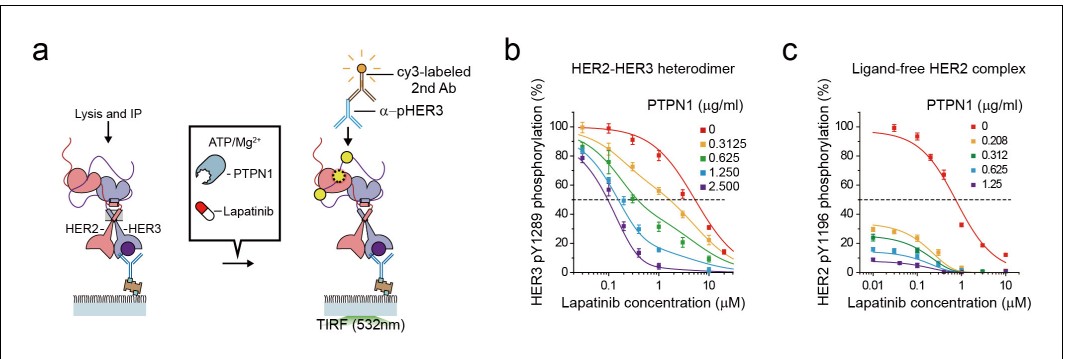

**Figure 7.** The high catalytic rate of the HER2-HER3 heterodimer is pivotal to its lapatinib resistance. (a) Schematics for inhibition of HER2-HER3 heterodimer with lapatinib under increasing concentration of PTPN1. (b, c) Lapatinib inhibition curves for HER2-HER3 heterodimers (b) and ligand-free HER2 complexes (c) obtained with increasing concentrations of PTPN1. Lapatinib, PTPN1, ATP and $Mg^{2+}$ were applied simultaneously. All data points and bar graphs were obtained from 10 different images (technical replicates; mean, SD).

heterodimers on live cell membranes by adding cognate ligands. After cell lysis, we induced rapid immunoprecipitation of these endogenously-formed heterodimers to the imaging surface of a single-molecule fluorescence microscope. After removing the cell extract via microfluidic buffer exchange, we examined the immunoprecipitated HER2-HER3 heterodimers using various biochemical tools at single-molecule resolution. These biochemical data provided unprecedented details as to how HER2 generates strong proliferative signals, in particular, when complexed with HER3. We expect that provided with high-affinity antibodies for pulling down other target membrane receptors, the methods we describe here will be generally applicable beyond the HER family receptors to other important signaling receptors, including those used by the human immune systems.

Notably, we confirmed that the immunoprecipitated HER2-HER3 heterodimers maintained their Tyr kinase activity. We found that the detergent used for cell lysis and immunoprecipitation is critical for the successful reconstitution of Tyr phosphorylation. Of the various detergents we tested, only digitonin, GDN, and GDN derivatives are able to preserve the Tyr kinase activity of the HER2-HER3 heterodimers. We further found that the different types of detergents can be exchanged but the concentration of each detergent has to remain above its respective CMC. Even a brief absence of detergent in the reaction buffer markedly impairs the Tyr kinase activity, suggesting irreversible structural changes are induced in the trans- and juxta-membrane domains during detergent depletion. GDN and digitonin have multiple glycosides and disogenin as the hydrophilic and hydrophobic groups, respectively. Comparison of this chemical structure with those of OG, DDM and CHAPSO, which failed in our function reconstitution of Tyr phosphorylation, suggests that the disogenin group plays an important role in reconstitution of the tyrosine kinase activity of the HER2 complexes. Presumably, the cholesterol-like structure of disogenin interacts with the membrane-associated domains of the HER2-HER3 heterodimers to critically modulate the tyrosine kinase activity.

In our reconstitution of Tyr kinase activity, we found that the kinase domains of HER2-HER3 heterodimers rigorously adopt the asymmetric dimer conformation. We further found that the HER3 kinase domain functions as the activator in the asymmetric dimer while the HER2 domain primarily functions as the receiver, consistent with the current prevailing view on the HER2-HER3 heterodimers. Because of these fixed roles, we initially presumed that the two kinase domains could be largely locked in the asymmetric dimer configuration with limited conformational fluctuations. Individual ligand-free HER2 complexes, on the contrary, have more than one HER2 kinase domains that likely play alternating roles as activator and receiver. We anticipated that the ligand-free HER2 complexes would experience substantial conformational fluctuations between active asymmetric dimer and 'non-dimer' conformations, in the latter of which the juxta-membrane and kinase domains are dissociated from one another.

Our single-molecule FRET measurements probing the JM-A domains indeed revealed that a substantial population (~50%) of the ligand-free HER2 complexes sample the non-dimer conformations, as reflected by the observed low FRET *E* values. Unexpectedly, a similar fraction (~35%) of the single

HER2-HER3 heterodimers showed the same low $E$ values, suggesting that their HER2 and HER3 kinase domains could be in largely dissociated states despite the fixed roles. In addition, when we measured the Michaelis-Menten curves, the single HER2-HER3 heterodimers and the ligand-free HER2 complexes exhibit highly similar $K_M$ values (29.2 and 16.5 µM). This result can be interpreted as the HER2-HER3 heterodimer and the ligand-free HER2 complexes spending similar fractions of time in the ATP-bound state at given ATP concentrations. Our observations collectively suggest that the kinase domains of HER2-HER3 heterodimers are highly dynamic, rather than being locked down, sampling the active dimer and the non-dimer conformations at a rate comparable to the ligand-free HER2 complexes.

How does the HER2-HER3 heterodimer then generate such persistent signaling despite the unexpected large conformational fluctuations in its cytoplasmic domains? The strong signaling potency of the HER2-HER3 heterodimer can be explained by its unique repertories of downstream effectors and kinase substrates. Indeed, a single HER2-HER3 heterodimer carries seven binding sites for PI3K and we observed simultaneous binding up to five copies of p85α to a single HER2-HER3 heterodimer. However, when we examined only the generation of a single type of pTyr residue, the HER2-HER3 heterodimer and the ligand-free HER2 complexes still behaved so differently, suggesting the possibility that they are distinct from one another even at the biochemistry level. Our Michaelis-Menten analysis revealed that it is the catalytic rate ($k_{cat,max}$) that exhibits the largest contrast between the HER2-HER3 heterodimer and the ligand-free HER2 complexes. This led us to hypothesize that despite substantial conformational fluctuations, the HER2 and HER3 kinase domains exhibit a remarkably high enzymatic potency once they form the active dimer conformation together. This high catalytic rate likely enables the HER2-HER3 heterodimer to outcompete the phosphate activity and the inhibitor binding, thereby continuously generating new pTyr residues. Indeed, in the presence of excessive Tyr phosphatases that reduce the effective catalytic rate, the HER2-HER3 heterodimer becomes susceptible to lapatinib inhibition, accompanied by a plunge in $IC_{50}$ by two orders of magnitude.

At the same time, we observed that the single HER2-HER3 heterodimers can physically interact with multiple copies of downstream interactors. In our time-resolved single-molecule fluorescence imaging, we found that binding of multiple interactors persists even under harsh dephosphorylating conditions designed to mimic the environments of cellular cytoplasm. Our observation suggests that while satisfying the conformational requirements for the Tyr kinase activity, the cytoplasmic domains of HER2-HER3 heterodimers exploit the remaining conformational flexibility to make multiple PPIs with downstream effectors. Owing to this ability to multitask, the newly generated pTyr residues can directly lead to recruitment of additional downstream signaling proteins with no serious latency. Therefore, our observations provide a telltale hint that the unusually high catalytic rate and the ability to multitask could be the key biochemical contributors to the persistent signaling potency of the HER2-HER3 heterodimers observed in the physiological milieu.

## Materials and methods

See *Supplementary file 1* for key resources used.

### Cell lines

Our HEK293T cell line was donated from Prof. Heo's lab. We purchased SKBR3 cell line from both Korean Cell Line Bank and ATCC. All the cell lines were re-authenticated through STR profiling (Macrogen STR service). Moreover, we checked mycoplasma contamination status using mycoplasma PCR detection kit (Lilif; Cat.# 25237) and we found our cell lines are mycoplasma free.

### Gene expression

We electroporated HEK293T or SKBR3 with two 35 ms pulses of 950 V to induce expression of eGFP- or mCherry-tagged proteins or other genes. HER3-eGFP was expressed in SKBR3 cells. After electroporation, we cultured the cells for 18–24 hr. In addition, we serum-starved the cells for 12–18 hr before harvesting them after treatment with a ligand (e.g., NRG1-β1). Unlabeled HER2, and HER3-eGFP (or unlabeled HER3 and HER2-eGFP) were expressed in HEK293T cells. Then, we cultured the cells for 18–24 hr, serum-starved them for 12–18 hr, and finally harvested them. Unlabeled (or YbbR-inserted) HER2 and HER3 were expressed in HEK293T cells. After electroporation, we

cultured the resulting cells for 18–24 hr. In addition, we serum-starved the cells for 12–18 hr before harvesting them after treatment with a ligand (e.g., NRG1-β1). Other eGFP- or mCherry-labeled proteins (i.e., PLCγ$_{SH2}$, p85α, and PI3K) were expressed in HEK293T cells. We cultured these cells too for 18–24 hr before harvesting. Finally, we divided the harvested cells into appropriate aliquots and stored them at −80℃ for up to 6 months as cell pellets.

## Cell lysis

All of our lysis buffers shared the same basic components—50 mM HEPES-NaOH (pH 7.4), 150 mM NaCl, 1 mM EDTA, 10% glycerol, and 2% protease inhibitor cocktail (Sigma Aldrich, P8340). According to experimental demands, we added additional detergents and a phosphatase inhibitor cocktail (Sigma Aldrich, P5726) to this base lysis buffer. We dissolved the detergents (Digitonin, GDN, GDN variants, DDM, OG, CHAPSO, and Triton-X-100) at 1–2% (w/v) for cell lysis. Because digitonin aggregates in solution within 8 hr at 4℃, we dissolved digitonin just before the lysis. After lysis, we centrifuged the cell lysates at 15000 g at 4℃ for 10 min and collected the supernatants for further analysis. We quantified the total amount of protein using a DC protein assay kit. Finally, we divided the supernatants into appropriate aliquots and stored them at −80℃ for up to 1–2 days.

## Preparation of SKBR3

We purchased SKBR3 from the Korean cell line bank and the American Type Culture Collection (ATCC). We cultured and maintained each breast cancer cell line in the media recommended by its distributor. From each 100 mm culture dish (~80% confluency), we obtained 400 μl of cell extract at 3–6 mg/ml total protein concentration. To harvest the cells, we removed the culture media and washed once with 37℃ DPBS. We then removed the DPBS and added 1 ml of 4℃ DPBS for cell resuspension. We detached the cells from the dish using a cell scraper and completely re-suspended them for aliquoting. Then, we centrifuged the re-suspended cells at 5,000 g for 5 min at 4℃ to remove any residual DPBS. We discarded this supernatant and stored the pellets at −80℃.

## Preparation of PEG coated glass surface

We prepared PEG-coated glass surfaces as previously reported (*Lee et al., 2013b*). We cleaned glass coverslips and quartz slides with piranha solution (2:1 = H$_2$SO$_4$:30% H$_2$O$_2$) for 30 min before washing them with distilled water (18 MΩ). To coat the glass surfaces with amine groups, we incubated the cleaned coverslips and slides with amino silane solution (3% N-[3-(Trimethoxysilyl))propyl] ethlenediamine, 10% acetic acid in methanol) for 20 min. We then washed the coverslips and slides with methanol and distilled water before drying them with nitrogen gas. After dissolving biotin-PEG and mPEG powder in 0.1 M sodium bicarbonate and 0.4 M potassium sulfate (biotin-PEG: mPEG ~3:100), we pushed the solution through a filter with 0.2 μm pores. We poured the PEG solution onto the slides and added the coverslips. Then, we washed the PEGylated quartz slides and coverslips with 18 MΩ distilled water, dried them with nitrogen gas, sealed them, and stored them at −20℃. To make the flow chambers, we thawed the PEGylated quartz slides and coverslips at room temperature for more than 5 min and assembled them using double-sided adhesive tape as a spacer between the layers. Then, we sealed the chamber with epoxy except at its inlet and outlet. We typically stored the mPEG-coated quartz slides and coverslips for up to 2 weeks before use. We found keeping the chambers in a food-saving vacuum container preserves their quality for over a month.

## Single-molecule fluorescence imaging

All the single-molecule fluorescence traces are obtained with home-built prism-Total Internal Reflection Fluorescence (TIRF) microscope. Single-molecule imaging to counting the number of molecules is proceeded using objective-TIRF microscope. Images and movies were excluded when unexpected large speckle due to cell debris in lysate, or defects during PEG-coated surface preparation were observed.

## Single-molecule immunoprecipitation of HER2 complexes on PEG-coated glass surfaces

We incubated the PEG-coated glass surfaces with neutravidin (0.1 mg/ml) for 5–10 min. After washing away the remaining neutravidin, we added anti-HER2 or HER3 antibodies (2–5 μg/ml) to the

surfaces and incubated them for 5–10 min. After washing the remaining antibodies, we applied cell lysates (0.1–1.0 mg/ml for ligand-free HER2 complexes and 1.0–4.0 mg/ml for HER2-HER3 heterodimers) to the surfaces and incubated them for 15 min. Finally, after washing the remaining lysates, we used the resulting surfaces for the rest of our experimental protocols.

## Phosphorylation of HER2 complexes

We used mainly NRG1-β1-treated or untreated SKBR3 cells to immunoprecipitate HER2 complexes on analysis chips. After treating the SKBR3 cells with lysis buffer containing 1% (w/v) digitonin or GDN, we diluted the resulting lysates to maintain the detergent concentration above its CMC (~1% (w/v) for digitonin,~0.1% (w/v) for GDN). We applied the lysates to PEG-coated surfaces also coated with HER2 or HER3 antibodies and then incubated them for 15 min. Then, we washed the remaining lysates with washing buffer (50 mM HEPES-NaOH (pH 7.4), 150 mM NaCl, 1% (w/v) glycerol, 0.1% (w/v) digitonin or 0.01% (w/v) GDN). Finally, we performed the phosphorylation of immunoprecipitated HER2 dimers by adding appropriate concentration of ATP and 10 mM $MgCl_2$ to the wash buffer. The phosphorylation reaction was terminated by removing ATP and $MgCl_2$ by introducing fresh washing buffer. We confirmed phosphorylation by applying phospho-tyrosine-specific antibodies with cy3 labeled 2nd antibody or eGFP-tagged downstream proteins.

## HER2- or HER3-YbbR cloning for coA-Cy3, coA-Cy5 labeling

To insert a linker-containing YbbR sequence (20 amino acids, 60 bp), we designed 65 bp oligonucleotides for PCR and assembly (HER2linkerybbRJM_F, HER2linkerybbRJM_R, HER3linkerybbRJM_F, HER3linkerybbRJM_R; See Key Resources Table at STAR METHODS section). We purchased these oligonucleotides from Macrogen, Inc Considering the length of the primer and its $T_m$, we performed the PCR reactions at annealing temperatures of 58℃ and 61℃. We designed the primer pairs to overlap by about 20 bp in the middle of the YbbR sequence.

## In vitro YbbR labeling and preparation for the single molecule fluorescence imaging

The final concentration for each reagent was as follows: 5 mg/ml of cell lysate containing HER2 or HER2 and HER3 ybbR, 10 µM for each coA-Dye, and 1 µM Sfp synthase in 50 mM HEPES buffer containing 10 mM $MgCl_2$, 1 mM DTT, and 1% (w/v) digitonin. We conducted the labeling reaction in a PCR tube with a final volume of 25 µl, at 37℃ for 1 hr. Then, we diluted the mixture 3x before applying it to a desalting column for free dye elimination. We incubated biotinylated anti-HER2 (for ligand-free complexes FRET) or anti-HER3 (for heterodimer FRET) antibodies with the neutravidin-treated biotinylated-PEG slides as previously described. To avoid non-specific dye binding, we washed the chambers with 1% (w/v) BSA each after neutravidin/antibody incubation. We then additionally diluted each reaction mixture by 15x, added them to the chamber, incubated them for 15 min, and then washed them with ATP and $MgCl_2$-containing imaging buffer. We used a final buffer containing 200 µM ATP and 10 mM $MgCl_2$ and 0.1% (w/v) digitonin (in 50 mM HEPES pH 7.4, 150 mM NaCl) to sustain kinetic activity. The imaging buffer we used for reducing the blinking and bleaching of dyes (especially CoA-Cy5) contained 1.5 mM Trolox, 2.5 mM PCA, and one unit/ml PCD.

## Single-molecule fluorescence traces for FRET calculation

We measured Cy3 and Cy5 fluorescence with a prism-type TIRF microscope that has a temporal resolution of 0.1 s/frame. The first few (~15) frames were excited with a 640 nm laser to check for the presence of Cy5 dyes. Then, the later frames were excited with a 532 nm laser for observations. The following scenarios were excluded: a) no Cy3 or Cy5 signal at either excitation, b) insufficient sum intensity (Cy3+Cy5) at the 532 nm excitation, c) Cy3:Cy5 stoichiometry obviously smaller or larger than 1:1. After this exclusion process, we manually checked and corrected the local backgrounds of each pair of FRET molecules. Also, we manually removed any blinking.

## FRET efficiency quantification

To ignore HER2 dimers that already lost their kinase activity, we had to exclude the traces which steadily showed low apparent FRET efficiency near 0.12 (the Cy3 leakage factor, beta). Considering

that our traces show relatively noisy data, we eliminated the traces which never show the apparent FRET efficiency over 0.4. Using the intensity of the Cy3 donor(I(d)) and the Cy5 acceptor(I(a)) of the selected traces, we calculated the apparent FRET efficiency for the $E$ histogram as follows: I(a)/ [I(d)+I(a)]. We calculated the corrected FRET efficiencies for the representative traces assuming Cy3 signal leakage (beta, around 0.12) as follows: [I(a)-beta*I(d)]/ [I(d)+I(a)]. Single-molecule traces were collected then donor-bleached frames were excluded. FRET efficiency value at each frame was calculated and collected for FRET efficiency distribution. Distribution was normalized by total population (total number of frame collected).

## HMM algorithm to find the number of probes and characterize binding kinetics

From images generated by selectively integrating pixels only where and when eGFP signals appeared, we selected ROIs (regions of interest) of 3 × 3 pixels each around local maximum pixels. We then recorded fluorescence traces from single ROIs and then removed the background by subtracting smoothed images (with 6 × 6 pixel kernels) from each frame. After scaling these traces from 0 to 1 (dividing by 800 (A.U), which is 10 times larger value then each single eGFP intensity value), we measured the BIC (Bayesian information criterion) values for various step numbers. Our method for calculating BIC values and the program we used to analyze the data were developed in a previous study (*Lee, 2009*). The program we used determines the BIC value along with a transition matrix and the position of each state using traces and a given number of states. We used the Origin program to fit the BIC values with piecewise linear functions for the purpose of determining the most likely number of states. BIC values were measured using representative traces from brightest spots (n ≥ 10) to find the maximum number of probes binding HER2 complex. After maximum number and position of state were measured, we applied the position of state as constant to find the number of states for the rest of inhomogeneous traces, assuming each Gaussian distribution only depend on number of probes binding. Because of the heterogeneity of each molecule, transition matrixes and standard deviations of each state were not fixed. We used the program described above to find the most likely state by fitting the piecewise step functions. We then used the fitted step functions to analyze the number of binding probes in each ROI and characterize their kinetics.

## Quantification and statistical analysis

Data are presented as mean ± standard deviation. All the bar graphs were obtained from 10 independent measurements unless obtained from model fitting, or otherwise stated.

## Acknowledgements

This work was supported by the National Creative Research Initiative Program (Center for Single-Molecule Systems Biology to T-YY; NRF-2011–0018352) and the Bio & Medical Technology Development Program (NRF-2018M3A9E2023523) funded by the National Research Foundation of South Korea.

## Additional information

### Competing interests

Byoungsan Choi: filed a patent on these findings (10-2018-0125506) and is now a senior scientist at Proteina. Dae Hee Lee, Seul Lee: is an employee at Proteina. Tae-Young Yoon: filed a patent on these findings (10-2018-0125506) and is co-founder of Proteina. The other authors declare that no competing interests exist.

### Funding

| Funder | Grant reference number | Author |
| --- | --- | --- |
| National Research Foundation of Korea | NRF-2011-0018352 | Tae-Young Yoon |

| National Research Foundation of Korea | NRF-2018M3A9E2023523 | Tae-Young Yoon Byoungsan Choi |

The funders had no role in study design, data collection and interpretation, or the decision to submit the work for publication.

## Author contributions
Byoungsan Choi, Conceptualization, Investigation, Visualization, Methodology, Writing - original draft, Writing - review and editing; Minkwon Cha, Conceptualization, Investigation, Methodology, Writing - review and editing; Gee Sung Eun, Conceptualization, Investigation, Writing - review and editing; Dae Hee Lee, Seul Lee, Investigation, Writing - review and editing; Muhammad Ehsan, Investigation; Pil Seok Chae, Won Do Heo, Resources; YongKeun Park, Supervision, Writing - review and editing; Tae-Young Yoon, Conceptualization, Supervision, Funding acquisition, Writing - original draft, Project administration, Writing - review and editing

## Author ORCIDs
Byoungsan Choi (iD) https://orcid.org/0000-0002-9679-4233
Minkwon Cha (iD) https://orcid.org/0000-0003-2316-8984
YongKeun Park (iD) http://orcid.org/0000-0003-0528-6661
Tae-Young Yoon (iD) https://orcid.org/0000-0002-5184-7725

## Decision letter and Author response
Decision letter https://doi.org/10.7554/eLife.53934.sa1
Author response https://doi.org/10.7554/eLife.53934.sa2

# Additional files

## Supplementary files
- Supplementary file 1. Key Resources Table.
- Transparent reporting form

## Data availability
All data generated or analysed during this study are included in the manuscript and supporting files.

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
