## [Decision Letter]

**Acceptance summary:**

The paper reports new insights into the biochemistry of HER2-HER3 heterodimers revealed by single-molecule analysis of receptor dimers immunoprecipitated from mammalian cells. By using fluorescent-based microscopic techniques, the authors characterized the stoichiometry, enzymatic activity, conformational dynamics and the downstream adaptor-binding capability of the heterodimer. As HER2-HER3 heterodimers play important roles in cancer development and metastasis, this method provides a different perspective for understanding the biology of ErbB receptors.

**Decision letter after peer review:**

Thank you for submitting your article "Single-molecule functional anatomy of endogenous HER2-HER3 heterodimers" for consideration by *eLife*. Your article has been reviewed by three peer reviewers, including Yibing Shan as the Reviewing Editor and Reviewer #1, and the evaluation has been overseen by Jonathan Cooper as the Senior Editor.

The reviewers have discussed the reviews with one another and the Reviewing Editor has drafted this decision to help you prepare a revised submission.

Summary:

In this manuscript, the authors described a single-molecule study of HER2-HER3 heterodimers immunoprecipitated from mammalian cells. By using fluorescent-based microscopic techniques, the authors characterized the stoichiometry, enzymatic activity, conformational dynamics and the downstream adaptor-binding capability of the heterodimer. As HER2-HER3 heterodimers play important roles in cancer development and metastasis, the method developed by the authors could potentially provide a different perspective for understanding the biology of ErbB receptors. While recognizing the potential of this study, the reviewers raised several important issues that need to be addressed.

Essential revisions:

1) It is not immediately clear why the HER2/HER3 heterodimers captured on TIRF coverslips using anti-HER3 antibodies, which are presumably bivalent, were predominantly single HER2/HER3 heterodimers. Wouldn't each antibody on the cover slip be expected to capture at least two HER2s or HER3s? Unlikely HER2 or HER3 binding to one of the Fab arms would preclude binding to the other? This issue is important because it raises a question about the reliability of the monomer/dimer/oligomer assignments. Couldn't the fact that the "majority" of IP'ed HER2 included two HER2 molecules simply arise from the bivalence of the capturing antibody?

2) It needs to be clarified whether the eGFP used in the experiments was the wild type, dimeric form or the mutated, monomeric form. If it were the wild type eGFP, the raw data of photobleaching experiments and validation of their claims on single step photobleaching should be included.

3) A potentially interesting and important result reported but not discussed by the authors is the observation of phosphorylation in the HER2 tail in what are reported as HER2/HER3 heterodimers. It has long been a question how the HER2 tail gets phosphorylated within a HER2/HER3 heterodimer in which phosphorylation is believed only to occur in trans (HER3 is a pseudokinase with negligible kinase activity). It has been suggested that HER2/HER3 pairs must form higher-order oligomers for the HER2 tail to be phosphorylated, and Landgraf and colleagues selected a HER3-aptamer that blocks HER2 but not HER3 phosphorylation and is thought to act by blocking formation of a putative higher-order oligomer. If the authors have indeed shown that the HER2 C-tail can be phosphorylated within an isolated HER2/HER3 heterodimer, I think that is an important result. However, if the number of HER2/HER3 heterodimers is not correctly determined (see point 1) this conclusion is obviously not justified.

4) The authors state that their measured kinetic parameters are much greater than previous parameters measured for EGFR. However, I think they are in line with previous measurements. The authors report apparent *K*_M_'s for ATP of the HER2 kinase in HER2/HER3 heterodimers as 16-30 μM and *k*_cat_'s in the 0.1-0.2/second range using the rate of appearance of fluorescence from antibodies recognizing pY in the C-tail. The authors observe a ~150-fold increase in *k*_cat_ when ligand is present. These values are very similar to those observed by Qiu et al. for EGFR (47 μM *K*_M_ and 0.4/second *k*_cat_) using purified EGFR and a peptide substrate and who observed a ~500-fold increase in *k*_cat_ when ligand is present. The authors estimate a ~12 fold higher catalytic rate (boosting their 0.22/second *k*_cat_ to 2.6/second) based on the presence of 12 tyrosines in the HER2/HER3 C-tails, but this assumption (i) seems flawed owing to potentially different *k*cat’s for different tyrosines, and (ii) renders the resulting *k*_cat_ not comparable to *k*cat’s determined with a single-tyrosine containing peptide. I find the author's ability to measure what appear to be reasonable kinetic parameters with this indirect method to be impressive and remarkable, but don't find the results to justify a conclusion of catalytic rates an order of magnitude higher than previously reported.

5) In the enzymatic assay (Figure 3), an anti-phosphotyrosine antibody and a Cy3-labelled secondary antibody were used for detecting the phosphorylation levels and the enzymatic kinetics of the heterodimer. Such quantification requires that the fluorescent intensity of bound Cy3 is linearly proportional to the tyrosine phosphorylation level. However, such an experimental calibration curve was missing in the manuscript.

6) The novelty of this report is the method and demonstration that it can be used to measure key biochemical and biophysical parameters of a key signaling and oncogenic ErbB oligomer. The authors tout the "unusually high [catalytic] rate" as underlying the stronger proliferative signal that has been observed for HER2/HER3 heterodimers, but this rationale is something of a straw man as the different signaling nature/strength of the HER2/HER3 heterodimer may just as easily explained by the different downstream effectors and kinase substrates of the HER2/HER3 pair. This needs to be addressed either by showing that the high catalytic rate is essential for the signaling potency of the HER2/HER3 pairing, or by correction of the related statements.

7) In the experiments, the authors found that HER2-HER3 heterodimer and ligand-free HER2 complex had similar *K*_M_ values but different *k*_cat_ values. Their interpretation was that these two complexes spend "similar fractions of time in the enzymatically competitive state at given ATP concentrations". But from the point of view of enzymatic kinetics, such results indicated that both complexes have similar ATP binding affinity or spend similar fractions of time in ATP-bound state at given ATP concentrations. ATP-bound state may or may not be equivalent to the enzymatically competitive state, which is also hard to be defined in single-molecule experiments.

8) To study the conformational dynamics of the heterodimer, the authors introduced an 11-residue-long tag into the JM region of the receptors and labelled the receptors with Cy3/Cy5. However, the authors didn't show that the labelled heterodimers had similar activity as the wild type heterodimers. In addition, as the receptors extracted/purified in detergent were not stable, it was not clear whether the conformational dynamics observed in the experiments reflects the equilibrium among different association states or the stability of the complexes. A comparison of FRET efficiencies at different time points after IP may give a better indication.

9) Is the dynamic nature of the HER2-HER3 dimers inherent in vivo or an artifact of the absence of membrane in the in vitro systems? It is possible and perhaps even likely that cell membrane may stabilize the juxta-membrane dimer and consequently the kinase dimer.

10) The finding that multiple effectors can bind with one HER2/HER3 dimer is important. Is the effector binding at different pTyr sites entirely stochastic or there is a pattern dictating the effector binding? Is one tyrosine phosphorylation particularly critical? This seems a good opportunity to answer such questions using mutagenesis.

---

## [Author Response]

Essential revisions:1) It is not immediately clear why the HER2/HER3 heterodimers captured on TIRF coverslips using anti-HER3 antibodies, which are presumably bivalent, were predominantly single HER2/HER3 heterodimers. Wouldn't each antibody on the cover slip be expected to capture at least two HER2s or HER3s? Unlikely HER2 or HER3 binding to one of the Fab arms would preclude binding to the other? This issue is important because it raises a question about the reliability of the monomer/dimer/oligomer assignments. Couldn't the fact that the "majority" of IP'ed HER2 included two HER2 molecules simply arise from the bivalence of the capturing antibody?

For all the surface IP cases we studied in this work, the total counts of the pulled down complexes were kept below 1000 in our imaging area of 40×80 μm^2^. This corresponds to a low surface density with a large inter-distance of more than 1.7 μm (on average).

Thus, although we used usual bivalent antibodies for our surface IP, we mainly captured a single HER2-HER3 heterodimer (or a single ligand-free HER2 complex) per antibody. This allowed us to observe mainly single photobleaching steps as in Figure 1D. We used this sparse pull-down condition throughout this work unless otherwise specified. In the revised manuscript, we have included the statement for the surface IP condition and added Figure 1—figure supplement 1D-G.

2) It needs to be clarified whether the eGFP used in the experiments was the wild type, dimeric form or the mutated, monomeric form. If it were the wild type eGFP, the raw data of photobleaching experiments and validation of their claims on single step photobleaching should be included.

We used the wild type eGFP for all our experiments. Because we used a low surface density for our experiment (see our response to Comment 1), we induced pull-down of ~1,000 complexes at the target concentration of ~1 nM (Author response image 1; with IP time typically being 15 minutes). Given that the known dissociation constant for dimerization of the WT eGFPs is 110 μM, which is five orders of magnitude larger than the concentration we used for surface IP, we estimate that the probability of seeing any dimerizing behavior of WT eGFPs is extremely low in our photobleaching analysis.

**Author response image 1. respfig1:** Immunoprecipitation of HER2 (at 1 nM), HER3 (at 1 nM), and eGFP (at 10 pM) using each HER2, HER3, and GFP antibodies.

3) A potentially interesting and important result reported but not discussed by the authors is the observation of phosphorylation in the HER2 tail in what are reported as HER2/HER3 heterodimers. It has long been a question how the HER2 tail gets phosphorylated within a HER2/HER3 heterodimer in which phosphorylation is believed only to occur in trans (HER3 is a pseudokinase with negligible kinase activity). It has been suggested that HER2/HER3 pairs must form higher-order oligomers for the HER2 tail to be phosphorylated, and Landgraf and colleagues selected a HER3-aptamer that blocks HER2 but not HER3 phosphorylation and is thought to act by blocking formation of a putative higher-order oligomer. If the authors have indeed shown that the HER2 C-tail can be phosphorylated within an isolated HER2/HER3 heterodimer, I think that is an important result. However, if the number of HER2/HER3 heterodimers is not correctly determined (see point 1) this conclusion is obviously not justified.

We like to thank the reviewers for this insightful comment. As we have detailed in our response to Comment 1, the probability for capturing two HER2-HER3 heterodimers with a single antibody is extremely low. In fact, even the co-localization of two HER2-HER3 heterodimers within a diffraction-limited spot (within ~500 nm) is quite unlikely to occur at the low surface density we used. Thus, the majority of the HER2-HER3 heterodimers used in our surface-IP experiments should be biochemically isolated from one another.

We have experimentally re-confirmed the active phosphorylation of the HER2 tail within these isolated HER2-HER3 heterodimers, which is now added as the main figure panels of Figure 2C and 2D in the revised manuscript. In particular, we found that our observed increases in the HER2 pTyr levels essentially requires the NRG1-β1 addition before the cell lysis, indicating that most of the pTyr residues have indeed been generated within the HER2-HER3 heterodimers.

Our observations support the hypothesis that a single HER2-HER3 heterodimer can phosphorylate its HER2 tail with a potency almost equal to that for the HER3 tail. This suggests an active *cis*-phosphorylation activity by the HER2 kinase in the HER2-HER3 heterodimer.

4) The authors state that their measured kinetic parameters are much greater than previous parameters measured for EGFR. However, I think they are in line with previous measurements. The authors report apparent K_M_'s for ATP of the HER2 kinase in HER2/HER3 heterodimers as 16-30 μM and k_cat_ 's in the 0.1-0.2/second range using the rate of appearance of fluorescence from antibodies recognizing pY in the C-tail. The authors observe a ~150-fold increase in k_cat_ when ligand is present. These values are very similar to those observed by Qiu et al. for EGFR (47 μM K_M_ and 0.4/second k_cat_) using purified EGFR and a peptide substrate and who observed a ~500-fold increase in k_cat_ when ligand is present. The authors estimate a ~12 fold higher catalytic rate (boosting their 0.22/second k_cat_ to 2.6/second) based on the presence of 12 tyrosines in the HER2/HER3 C-tails, but this assumption (i) seems flawed owing to potentially different k_cat_ ‘s for different tyrosines, and (ii) renders the resulting k_cat_ not comparable to k_cat_ ‘s determined with a single-tyrosine containing peptide. I find the author's ability to measure what appear to be reasonable kinetic parameters with this indirect method to be impressive and remarkable, but don't find the results to justify a conclusion of catalytic rates an order of magnitude higher than previously reported.

In the previous version of the manuscript, to estimate the total catalytic rate of a single HER2-HER3 heterodimer, we multiplied the *k*_cat,max_ measured for a single Tyr residue (Y1289) by the total number of Tyr residues.

We understand the reviewers’ concerns and agree that this analysis could be misleading. We provide evidence that the *k*_cat_ for phosphorylation of other Tyr residues is in a similar order to that of pY1289 (e.g., the data points for pY1197, pY1222, pY1276 and pY1328 in Figure 3C and the corresponding data points in Figure 3D). However, we note that these observations do not directly lead to the conclusion that all these pTyr residues are generated to the same extents in each HER2-HER3 heterodimer. It is possible that only a subset of the Tyr residues is generated in a given HER2-HER3 heterodimer while the collection of these HER2-HER3 heterodimers shows the similar degrees of phosphorylation for different Tyr residues as a whole group.

Perceiving the potential pitfall in our analysis, we have removed the relevant analysis and the sentence in Discussion. In the revised manuscript, we have instead added a new set of data showing the high *k*_cat_ would be pivotal to the persistent signaling potency of the HER2-HER3 heterodimer as well as to its resistance against lapatinib (see our response to Comment 6 for details).

5) In the enzymatic assay (Figure 3), an anti-phosphotyrosine antibody and a Cy3-labelled secondary antibody were used for detecting the phosphorylation levels and the enzymatic kinetics of the heterodimer. Such quantification requires that the fluorescent intensity of bound Cy3 is linearly proportional to the tyrosine phosphorylation level. However, such an experimental calibration curve was missing in the manuscript.

Following the reviewer comment, we examined the linearity in our single-molecule immunolabeling with pTyr-specific antibodies. To increase the amount of surface pTyr residues in a linear fashion, we increased the total protein concentration of the SKBR3 cell lysate used for the surface IP. Under our sparse pull-down condition, the amount of the immunoprecipitated HER2 dimers or complexes linearly increased (as we show in Figure 1—figure supplement 1E, G).

We then carried out our immunolabeling for pTyr1289 (in the HER3 tail) and pTyr1196 (in the HER2 tail), respectively. We found that both immunolabeling counts indeed increased linearly with the total protein concentration. These data show the linearity of our assay, which in turn validate our determination of *k*_cat_ and *K*_M_ in Figures 3C and 3D. These new data sets are included as Figures 3—figure supplement 1A,B in the revised manuscript.

6) The novelty of this report is the method and demonstration that it can be used to measure key biochemical and biophysical parameters of a key signaling and oncogenic ErbB oligomer. The authors tout the "unusually high [catalytic] rate" as underlying the stronger proliferative signal that has been observed for HER2/HER3 heterodimers, but this rationale is something of a straw man as the different signaling nature/strength of the HER2/HER3 heterodimer may just as easily explained by the different downstream effectors and kinase substrates of the HER2/HER3 pair. This needs to be addressed either by showing that the high catalytic rate is essential for the signaling potency of the HER2/HER3 pairing, or by correction of the related statements.

We agree with the reviewer’s point that the strong proliferative signal of the HER2-HER3 heterodimer can be explained by its different repertories of downstream effectors and kinase substrates. Indeed, a single HER2-HER3 heterodimer carries seven binding sites for PI3K and we observed simultaneous binding up to five copies of p85α to a single HER2-HER3 heterodimer (e.g., Figure 5E). We have edited the Discussion section to explicitly state this fact in the revised manuscript.

Yet, when we examined only the generation of a single type of pTyr residue, the HER2-HER3 heterodimer and the ligand-free HER2 complexes still behaved so differently with the former being much more resilient than the latter in the Tyr kinase activity. This suggests that they are distinct from one another even at the biochemistry level (for example, Figure 3C versus 3D). To better understand this persistent biochemical behavior, we designed a new set of experiments in which we added HER2-targeting TKI, lapatinib, to inhibit the reconstituted Tyr kinase activity. We measured the degree of inhibition as a gauge of the signaling potency of the HER2 oligomers (i.e., the stronger the signaling activity is, the lower the inhibition would be at a given lapatinib concentration). We determined the lapatinib inhibition curves for the HER2-HER3 heterodimer and the ligand-free HER2 complexes, respectively and found that the IC_50_ values are 10 times higher for the HER2-HER3 heterodimer, likely reflecting its known strong resistance against the lapatinib inhibition (Figures 7B and 7C; red curves).

To examine the role of the high catalytic rate in the observed persistent biochemical behaviors, we added increasing concentrations of PTPN1 and re-examined the lapatinib inhibition curves. Remarkably, we found that the presence of PTPN1 markedly shifts the lapatinib inhibition curve of the HER2-HER3 heterodimer to a lower concentration range, up to two orders of magnitude at the highest PTPN1 concentration we studied (2.5 μg/ml). We reasoned that these PTPN1s in excess rapidly remove the generated pTyr residues and are thus expected to reduce the net effective catalytic rate, but without affecting the conformational changes of the HER2 oligomers (for example, those observed in Figure 4). Our results then suggest that as the catalytic rate wanes down, the HER2-HER3 heterodimer becomes more vulnerable to the lapatinib inhibition.

Our new observations seem to corroborate our hypothesis that the high catalytic rate makes a substantial contribution to the strong signaling potency of the HER2-HER3 heterodimer. Indeed, the ligand-free HER2 complexes that have a much lower catalytic rate instantly loses their signaling potency upon small addition of PTPN1 (Figure 7C). We have included these new observations in the revised manuscript as the main Figure 7.

7) In the experiments, the authors found that HER2-HER3 heterodimer and ligand-free HER2 complex had similar K_M_ values but different k_cat_ values. Their interpretation was that these two complexes spend "similar fractions of time in the enzymatically competitive state at given ATP concentrations". But from the point of view of enzymatic kinetics, such results indicated that both complexes have similar ATP binding affinity or spend similar fractions of time in ATP-bound state at given ATP concentrations. ATP-bound state may or may not be equivalent to the enzymatically competitive state, which is also hard to be defined in single-molecule experiments.

We thank the reviewer for pointing this out. We edited the corresponding words as follows:

“This result can be interpreted as the HER2-HER3 heterodimer and the ligand-free HER2 complexes spending similar fractions of time in the ATP-bound state at given ATP concentrations.”

8) To study the conformational dynamics of the heterodimer, the authors introduced an 11-residue-long tag into the JM region of the receptors and labelled the receptors with Cy3/Cy5. However, the authors didn't show that the labelled heterodimers had similar activity as the wild type heterodimers. In addition, as the receptors extracted/purified in detergent were not stable, it was not clear whether the conformational dynamics observed in the experiments reflects the equilibrium among different association states or the stability of the complexes. A comparison of FRET efficiencies at different time points after IP may give a better indication.

Following the reviewer comment, we determined the catalytic rates of the YbbR sequence-inserted HER2 oligomers and compared the values with those of the WT ones. We found that the YbbR-inserted oligomers maintained about 40% of the enzymatic activity for both HER2-HER3 heterodimer and ligand-free HER2 complexes (included as new Figure 4B in the revised manuscript). The large contrast between these two types of the HER2 oligomers was also maintained after the YbbR insertion. We examined a myriad of positions in the HER2 and HER3 kinase domains for the YbbR insertion. In fact, we found that only the positions reported in this work maintained the Tyr kinase activity to a reasonable extent. Although the YbbR-inserted HER2 oligomers may not be perfectly matching with the WT oligomers in terms of the enzymatic activity, the data presented in this work could provide unique insights into the conformational changes occurring in the cytoplasmic domains of HER2 and HER3.

We agree with the reviewer’s concern that the different FRET populations (in particular, low FRET populations) simply reflect some degraded HER2 oligomers, rather than different microstate sampling in the equilibrium. It is noteworthy that the incubation of the HER2 oligomers with lapatinib selectively enhanced the low FRET populations (Figure 4C and 4E). The observed FRET changes upon a biochemical treatment per se reflect conformational changes occurring within the HER2 oligomers.

In the revised manuscript, we stated the possibility that some of the low FRET populations represent degraded species as well as toned down our relevant statement.

“We note the possibility that different FRET populations, especially, those with low *E* values, reflect some degraded HER2 dimers or complexes. In addition, it is possible that solubilization of the HER2 dimers in micelles renders the low FRET populations (with JM-As dissociated) overestimated compared with the physiological populations in the cellular membranes. However, the FRET changes in response to the lapatinib incubation per se report detailed conformational changes in the single HER2 dimers. They also suggest that binding of lapatinib to the non-active conformation of the HER2 kinase domains leads to total dissociation of the asymmetric dimer of kinase domains.”

9) Is the dynamic nature of the HER2-HER3 dimers inherent in vivo or an artifact of the absence of membrane in the in vitro systems? It is possible and perhaps even likely that cell membrane may stabilize the juxta-membrane dimer and consequently the kinase dimer.

As discussed in our response to Comment 8, we cannot rule out the possibility that some low FRET populations reflect degraded HER2 oligomers. At the same time, as the reviewer points out, the observed broad FRET distributions, with many of the HER2 oligomers being in the JM-A dissociated state, may be an artifact of our in vitro pull-down experiments.

As we report in Figure 2, the detergent concentrations should always be kept above their respective CMCs to preserve the Tyr kinase activity of the single HER2-HER3 heterodimers. Even brief depletion of detergents largely abolishes the Tyr kinase activity. These observations indicate that the immunoprecipitated HER2-HER3 heterodimers are solubilized in detergent micelles, rather than exposed to the aqueous environment.

Yet, the membrane proteins experience much higher lateral pressure in the lipid bilayers than they do in micelles. There is a possibility that the low FRET populations are overestimated in our in vitro experiment when compared with the physiological populations found in the cellular membranes. In the revised manuscript, we also included this point in our discussion of the FRET data.

10) The finding that multiple effectors can bind with one HER2/HER3 dimer is important. Is the effector binding at different pTyr sites entirely stochastic or there is a pattern dictating the effector binding? Is one tyrosine phosphorylation particularly critical? This seems a good opportunity to answer such questions using mutagenesis.

This is a very interesting question. Inspired by the reviewer comment, we have introduced the Y-to-F mutation to six different Tyr residues in the HER3 tail using our isogenic expression system.

We reasoned that if there were any patterns dictating the orders of effector binding, we would see marked drops in the binding counts in certain mutants. This is because with no pTyr generated in the specific Tyr residue, we could prevent any further bindings occurring in a predefined order (i.e., the pattern in the effector binding).

We collected a huge set of time-resolved traces for the six mutants (total ~25,000 traces), and carried out analysis of identifying the number of p85α binding for individual traces. Unfortunately, we failed to observe any such marked changes in the p85α binding counts across the six different Y-to-F mutants we examined. All the mutants showed only slight decreases in the p85α binding counts. In fact, we found that virtually, every peak in the count distribution made one step to the left, thus a binding count decreased by exactly one on average. These data, now included as the main Figure 5H-M, seem to indicate that the effector binding largely occurs in a stochastic manner with no specific pattern dictating the order of effector binding, at least, in our in vitro setting.